# Multiparametric biophysical profiling of red blood cells in malaria infection

Shreya S. Deshmukh [1,2], Bikash Shakya [3], Anna Chen[4], Naside Gozde Durmus[5], Bryan Greenhouse [4], Elizabeth S. Egan [3] & Utkan Demirci [2✉]

Biophysical separation promises label-free, less-invasive methods to manipulate the diverse properties of live cells, such as density, magnetic susceptibility, and morphological characteristics. However, some cellular changes are so minute that they are undetectable by current methods. We developed a multiparametric cell-separation approach to profile cells with simultaneously changing density and magnetic susceptibility. We demonstrated this approach with the natural biophysical phenomenon of *Plasmodium falciparum* infection, which modifies its host erythrocyte by simultaneously decreasing density and increasing magnetic susceptibility. Current approaches have used these properties separately to isolate later-stage infected cells, but not in combination. We present biophysical separation of infected erythrocytes by balancing gravitational and magnetic forces to differentiate infected cell stages, including early stages for the first time, using magnetic levitation. We quantified height distributions of erythrocyte populations—27 ring-stage synchronized samples and 35 uninfected controls—and quantified their unique biophysical signatures. This platform can thus enable multidimensional biophysical measurements on unique cell types.

[1] Department of Bioengineering, Stanford University Schools of Engineering and Medicine, Stanford, CA, USA. [2] Canary Center for Early Cancer Detection, Department of Radiology, Stanford University School of Medicine, Palo Alto, CA, USA. [3] Department of Pediatrics; Department of Microbiology and Immunology, Stanford University School of Medicine, Stanford, CA, USA. [4] Department of Medicine, University of California San Francisco, San Francisco, CA, USA. [5] Department of Radiology, Stanford University School of Medicine, Palo Alto, CA, USA. ✉email: utkan@stanford.edu

Cells form the subunits of life and encode the physical diversity in life forms[1]. These unique physical signatures can map cells to different states: whether stages in the cell cycle[2,3], or on the spectrum from healthy to pathological (by infection[4–6], cancer[7,8], aging[9,10], pharmacological response[11,12], or other phenomena[13–15]). Membrane structure and porosity[16], or cell hydration[17], can be measured by fluorescent dye uptake[18], and is linked to lysis susceptibility, dehydration[19], cell health and apoptotic status[20], or treatment response[8]. Chemical composition, including cytoskeletal structure[21] and lipid content[22], also measurable by fluorescent microscopy, are fundamental properties of cells that are linked to their functions as well as disease outcomes. Density is a measure of cellular contents and their concentration, and can be manipulated in centrifugation, density gradient separation[23], etc.[24] but is also measurable by label-free magnetic levitation[25,26]. Density is linked to cells' metabolic states and relative hydration, lipid, and protein contents[27], with levitation being applied for live/dead cell separation[28] as well as detection of specific cell types, e.g., adipocytes[29]. Cellular magnetic susceptibility is a biophysical property linked to iron content (e.g., erythrocytes or red blood cells(RBCs))[30,31] or radical species (e.g., reactive oxygen species) that generate paramagnetic signatures[32]. RBCs are relatively simple cells lacking nuclei, with a high iron content in hemoglobin. Thus, magnetic susceptibility acts as an indicator of RBC state, varying with spin state due to oxygenation[33], or phenomena like infection[34]. Magnetic susceptibility differences can be manipulated in magnetic field gradients and manipulated indirectly by labeling cells with magnetic nanoparticles or beads[35]. Other morphological and rheological characteristics, such as deformability have also aided our understanding of mechanical properties linked to cellular health, prompting the development of new analysis techniques[36].

Such biophysical measurement methods have been instrumental in characterizing cellular properties, investigating unknown structure-function relationships, and charting the phenotypic manifestations of many pathologies. Malaria infection[37] offers a unique opportunity to study gradual changes in density and magnetic susceptibility occurring simultaneously. As the parasites grow within their host RBCs, they progressively digest hemoglobin. This process increases the magnetic susceptibility of infected RBCs via a buildup of paramagnetic hemozoin[38,39], while reducing the cells' overall density via hemoglobin depletion[6,40]. Magnetic susceptibility or density, independently, are commonly used to separate infected RBCs from uninfected RBC in P. falciparum cultures, for example with magnetic columns or density gradient centrifugation, respectively[41,42]. However, the early stages in the parasite's life cycle (called rings) are the prevalent stage in circulation, and have traditionally been difficult to separate from uninfected RBCs since the biophysical changes are still minute[38]. Further, the existing methods for enriching malaria parasites share a limitation with many of the biophysical measurements described above; they are typically limited to laboratory settings due to infrastructural dependence, and thus cannot be as widely applied in resource-limited settings.

We demonstrate here a magnetic levitation platform that leverages subtle biophysical differences in the inherent magnetic susceptibility and density signatures of blood cells to map a multidimensional space to the single metric of levitation height. We use levitation height as an inherently label-free, sensitive marker to link human blood cells to a range of normal to pathological conditions using malaria as an example[25,43–45]. While magnetic levitation has previously been used for various density measurements[22,46,47], this is the first time to our knowledge that magnetic levitation has been used to separate cells with a combination of density differences and magnetic susceptibility

differences. Here, we present a system that is simple to operate and could support sensitive biophysical measurements of cells in various operative settings. Using P. falciparum-infected erythrocytes as a case study, we profile these characteristics simultaneously by separation of infected cells by levitation height. With this method, we are able to distinguish not only late-stage infected RBCs but also the younger ring-stage cells which have a subtler biophysical signature. The unique levitation height distribution of each blood sample is imaged for quantitative analysis to subsequently identify the characteristic presence of infected cells. Since the mechanism of biophysical separation is power-free, this platform has the potential to be adapted for broader use—such as the resource-constrained settings where malaria is endemic—and could therefore enable diverse applications in biophysical cell separation.

## Results

**Magnetic levitation system design**. In developing an efficient and noninvasive method for biophysical measurement of cells, we performed magnetic levitation of heterogeneous cell populations, using levitation height as one metric to represent a combination of density and magnetic susceptibility. We used a compact device that suspends cells in a paramagnetic solution within a magnetic field, and imaged their height distribution, as in Fig. 1a. This system has previously been used to separate various cell types, such as cancer cells from leukocytes[25]. In this study, we aimed to show the separation of erythrocytes (RBCs) with different properties, using P. falciparum infection as a natural model of RBCs which simultaneously change in density and magnetic susceptibility. Cells levitate in a liquid suspension, adjusting vertical height positions over time as a function of their density and magnetic susceptibility with respect to the surrounding medium, until forces are balanced as shown in Fig. 1b. The physics of this force balance is described in Eq. 1 in the "Methods" section. To understand the cells' biophysical profile, we can snapshot them in their equilibrium positions to record their final vertical locations (without inducing flow). At this equilibrium point, typically achieved within 15 minutes, the height distribution of the cell population—the "levitation pattern"—is imaged and analyzed to build a quantitative profile of the cell population's biophysical characteristics, as in Fig. 1c. The information in these imaged patterns is then extracted as described in "Imaging of cell levitation distributions" in the Methods, to be subsequently analyzed.

**Mechanism of cell separation in levitation**. Individual cells suspended in the fluid of the levitation chamber have unique responses to a range of forces based on their inherent biophysical properties, as described in Fig. 1b. At equilibrium, the main forces affecting the vertical force balance in this system are gravity, buoyancy, and magnetic repulsion/attraction. The relationship between the density of the cell and the surrounding fluid affects the balance of gravity and buoyancy, whereas the relationship between the magnetic susceptibility of the cell and the fluid affects its position with respect to the magnetic field. The two main biological variables that influence the height ($h$) of the cell, are thus its density ($\rho$) and its magnetic susceptibility ($\chi$). Our work shows the first known demonstration of cell types with different $\rho$ and $\chi$ each levitating at distinct $h$ within the same system, as a function of both properties. While separation of cells with different $\rho$ has been demonstrated previously[25,26], this is also the first known demonstration of cell separation by $\chi$ in levitation. We used colloidal silica particles to increase and control the medium density and thus its buoyancy, as in Fig. 2a.

RBCs are primarily comprised of hemoglobin (96% of dry weight), an iron-based molecule that exhibits different magnetic

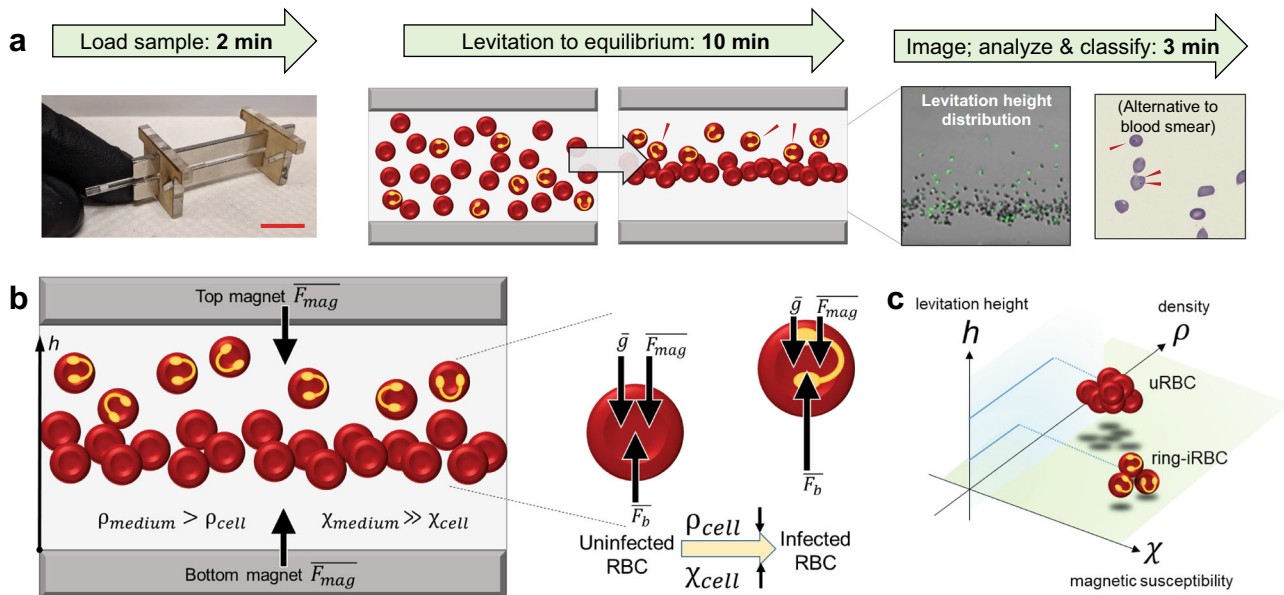

**Fig. 1 Mechanism of magnetic levitation for erythrocytes (RBCs).** h refers to levitation height (μm), X refers to magnetic susceptibility (unitless), and ρ refers to density (g/mL). RBC refers to red blood cells, or erythrocytes, uRBC to uninfected RBCs, ring-iRB to ring-stage synchronized infected RBCs, and troph-iRBC to trophozoite-stage (i.e., more mature) infected RBCs. **a** Workflow: First, a volume of blood (smaller than a fingerprick) is mixed with a paramagnetic solution within a glass capillary and loaded into the magnetic device. Scale bar = 1 cm. Second, cells begin to levitate at different heights, arriving at equilibrium within 12–16 min and resulting in height-based separation of different cell types. Third, cell levitation patterns in the device are imaged upon equilibrium using a microscope for analysis. Captured images are analyzed to produce levitation height distributions. Red arrows mark ring-stage infected RBCs in the schematic and images. **b** Schematic of device containing a *Plasmodium falciparum*-infected blood sample. RBCs (slightly less dense than the medium) levitate at equilibrium in the upper half of the channel (due to buoyancy and gravity), and, being relatively diamagnetic compared to the paramagnetic medium, away from the region of high magnetic gradient near the top magnet (due to the repulsive magnetic force). Infected blood samples have some RBCs containing ring-forms of the parasite, which are hypothesized to decrease the density and increase the magnetic susceptibility of the host RBC, thereby increasing buoyancy and decreasing the effects of gravity and magnetic repulsion. Thus, parasite-containing RBCs are hypothesized to levitate higher (z-axis) than uninfected RBCs. **c** As this 3-dimensional plot schematic shows, levitation height h can be used as a single metric to represent the multiparametric biophysical profile of single cells—in this case, density and magnetic susceptibility as they change in *P. falciparum*-infected cells.

susceptibilities depending on its electron spin state[48]. RBCs are therefore a prominent example of cells that contain naturally varying magnetic signatures, changing by oxygenation state, and inherently linked to pathological states[33,49]. This can be affected by anemia, genetic disorders, such as sickle cell anemia, and infection[19,50]. Malaria infection caused by *P. falciparum*, the deadliest form of the parasite, is the primary focus of this work. It is a naturally occurring biological example of cells undergoing multiple biophysical changes that represent infection stages, and can be harnessed for noninvasive manipulation and measurements, as we demonstrate. However, before testing the effect of multiple simultaneous biophysical changes as in malaria infection, we tested the concept by independently changing one cellular variable at a time using mechanisms independent to malaria.

First, we modified only the magnetic properties of RBCs to investigate the previously untested capability of this levitation method to separate cells on the basis of magnetic susceptibility ($\chi$). To modulate $\chi$ independently, we treated RBCs with sodium nitrite ($NaNO_2$) which is known to convert oxyhemoglobin (diamagnetic, i.e., no magnetic properties) into methemoglobin (paramagnetic, i.e., weakly magnetic within a magnetic field)[31]. As we show in Fig. 2b, $NaNO_2$-treated (paramagnetic) RBCs levitate towards areas of higher magnetic gradient more than untreated control (diamagnetic) RBCs do. In low-density medium (where the cells levitate closer to the bottom magnet), the closest area of high magnetic gradient is the bottom of the chamber, and the treated cells levitate lower than control cells, as expected. In high-density medium (when the cells levitate closer to the top

magnet), the closest area of high magnetic gradient is the top of the chamber, and the treated cells levitate higher than control cells, as expected. However, this difference is smaller than in the low-density case; we hypothesize that this is because the baseline height of the cells is much closer to the middle of the chamber, and therefore further from the top magnet. Overall, this suggests that cells' levitation heights can vary as a function of $\chi$ alone, at least in the low-density case.

Subsequently, we aimed to verify whether we could separate cells of differing density by levitation height, specifically with RBCs. To do this, we modulated cell $\rho$ alone by placing RBCs in solutions of different tonicity (but the same density) to change their density by osmosis. In a hypertonic sodium chloride (NaCl) solution, RBCs became denser due to dehydration and accordingly had a lower levitation height than in an isotonic control. In a hypotonic NaCl solution, RBCs became swollen with water and less dense, and accordingly had a higher levitation height than in an isotonic control. Overall, as we show in Fig. 2c, the cells' levitation heights changed as expected according to osmosis.

Together, these experiments show that $\chi$ and $\rho$ can be modulated independently in RBCs to produce corresponding levitation height changes; due to oxygenation as we show in Fig. 2b, and due to osmosis as we show in Fig. 2c. These experiments provided the basis to test cells in which both of these properties change together, as with *P. falciparum*-infected RBCs. This infection is known to progressively decrease $\rho$ and progressively increase $\chi$ in RBCs throughout the stages of infection as a direct consequence of the parasite's metabolic activities, which include aggressive consumption of the

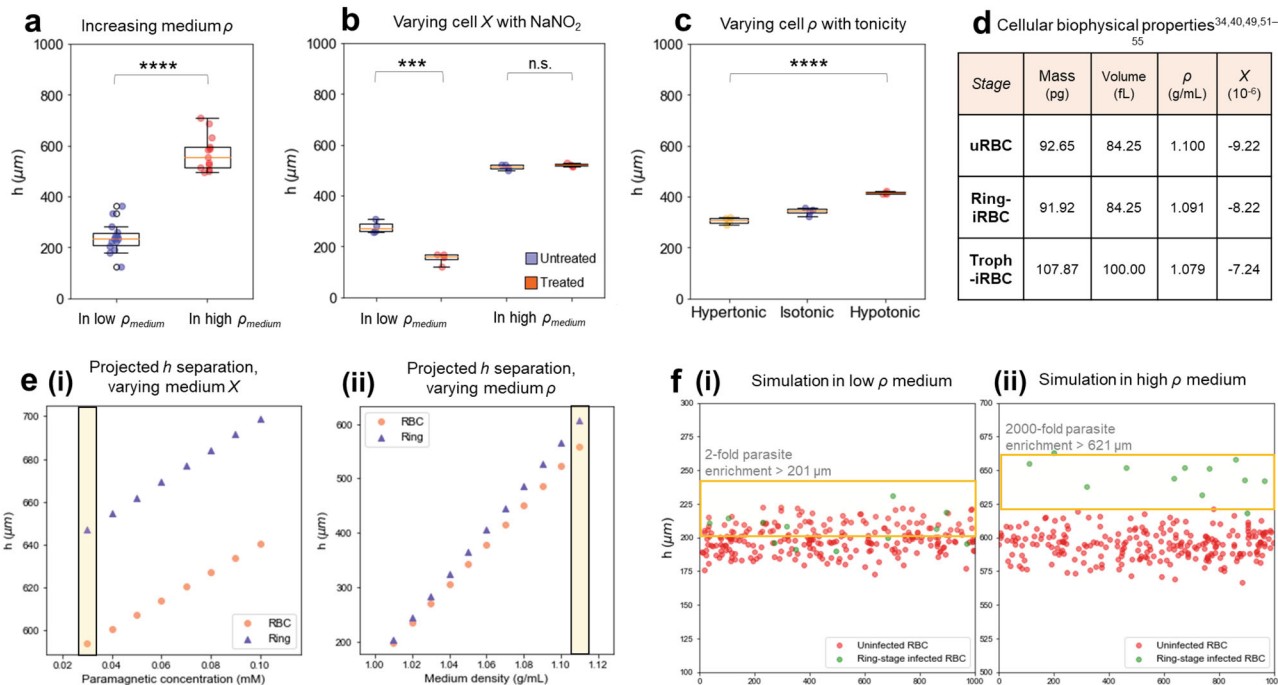

**Fig. 2 Levitation changes from independent variations in density and magnetic susceptibility, and theoretical predictions of levitation heights of cell types under different conditions, based on reported and estimated values of cell densities and magnetic susceptibilities.** $h$ refers to levitation height (μm), $X$ refers to magnetic susceptibility (unitless), and $\rho$ refers to density (g/mL). RBC to red blood cells, or erythrocytes, uRBC refers to uninfected RBCs, ring-iRBC to ring-stage synchronized infected RBCs, and troph-iRBC to trophozoite-stage infected RBCs. **a** Increasing the density ($\rho$) of the medium leads to increased levitation heights for the same cell type (uRBC). Low medium $\rho$ data are in purple circles ($n = 14$ independent samples), and high medium $\rho$ in orange circles ($n = 13$ independent samples), with $p = 2.4 \times 10^{-12}$. **b** uRBCs in levitation without treatment (purple circles) or with sodium nitrite (NaNO$_2$) treatment (orange circles) which converts the hemoglobin into the more paramagnetic methemoglobin. The experiment was conducted in low-density medium ($n = 4$ distinct RBC population samples from donor blood aliquots in the untreated condition, and $n = 4$ in the treated condition) with $p = 0.00029$, and in high-$\rho$ medium ($n = 4$ untreated, and $n = 4$ treated) with $p = 0.30$. **c** Uninfected red blood cells in levitation, in solutions of varying tonicity and controlled $\rho$. Cells became more or less dense due to osmosis, with cells in hypertonic solution ($n = 3$, yellow circles) becoming dehydrated and denser than in isotonic solution ($n = 3$, purple circles), and cells in hypotonic solution ($n = 3$, orange circles) becoming swollen and less dense than in isotonic solution, and their levitation heights behaving accordingly, with $p = 9.6 \times 10^{-7}$. **d** Biophysical characteristics of RBCs in various stages of *P. falciparum* infection, as reported or estimated from measurements in literature (see Fig. S1 for more). **e**(i) Predicted heights of ring-iRBCs (purple triangles) and uRBCs (orange circles) at varying medium magnetic susceptibilities (modulated via the paramagnetic ion concentration (calculated at high medium $\rho = 1.11$ g/ mL)). The selected paramagnetic concentration is marked in a yellow box. (ii) Predicted heights of ring-iRBCs (purple triangles) and uRBCs (orange circles) at varying medium densities, showing increased separation in a high-$\rho$ medium, marked in a yellow box (calculated at low medium $X = 1.12 \times 10^4$). **f** Simulation of ring-stage synchronized infected culture at 5% parasitemia and diluted 1:100, (i) in a low-$\rho$ medium (1.01 g/mL) at low $X$ (1.12 × 10$^4$), where ring-iRBCs (green circles) are enriched twofold above $h = 201$ μm (midpoint of the two cell types' projected heights); (ii) in a high-$\rho$ medium (1.11 g/mL) at low $X$ (1.12 × 10$^4$), where the concentration of ring-iRBCs is enriched 2000-fold above $h = 621$ μm (midpoint of the two cell types' projected heights). uRBCs are shown as red circles. In **a**, **b**, Welch's $t$-test was used to compare between groups. In **c**, ANOVA was used to compare across groups (asterisks indicate statistical significance across groups). *$p < 0.05$; **$p < 0.01$; ***$p < 0.001$; ****$p < 0.0001$. Box plots represent upper and lower quartiles, with the center line marking the median, and the whiskers showing the range of the data. All experimentally measured samples are biologically independent.

hemoglobin (the cell's primary content). These biophysical changes, documented in Fig. 2d, have been studied with a range of methods and replicated throughout the literature, with further details in Fig. S1 showing estimated biophysical changes in infection based on models from the literature[34,40,49,51–55]. The motivation to combine both biophysical properties in one measurement is to improve sensitivity by combining them to enhance overall separation. This is necessary for separating ring-stage infected RBCs, in which these biophysical changes are less pronounced than in later stages, making them difficult to separate by density or magnetic susceptibility alone.

**Modeling cell heights as a function of biophysical variables.** We modeled our system with variables that describe the changing cellular properties as well as the changing device parameters, to understand the relationship between the system's components

and to optimize cellular separation. Each cell moves through the fluid, at a rate dependent on its volume and the fluid's viscosity, until it reaches an equilibrium position where forces are balanced, and the only movement is Brownian motion. At the equilibrium height, the cell's height is a function of the interaction between buoyancy and the weak magnetic repulsion between the cell and the surrounding paramagnetic fluid. Thus, assuming equilibrium positions are reached, this model takes into account known parameters of the cell, fluid, and device dimensions[47]. This model predicts levitation height as a function of the force balance equations described in the "Methods" section that explain these relationships. In Fig. S2a, we show the theoretical levitation heights of uninfected RBCs and infected RBCs at different stages of infection, calculated using Eq. 1 with known or estimated density and magnetic susceptibility values from the literature (Fig. 2d and Fig. S1)[40,55,56]. We used these equations to predict cell levitation heights under varying fluid conditions (density or

magnetic susceptibility) as in Fig. 2e, further described in the "Methods" section. By predicting the heights of different cell types under a range of conditions, we explored which conditions would maximize the separation in height between uninfected RBCs and ring-stage infected RBCs, as in Fig. S2b. This model informed our optimization of the conditions required for maximal separation between uninfected RBCs and ring-stage infected RBCs. These two cell types are of primary interest for separation because they constitute the vast majority of a patient's RBC population due to sequestration of later stages, and because separation of later-stage infected RBCs is already demonstrated[38,42,57].

We show in Fig. 2e(i) that increasing the medium's magnetic susceptibility increases the overall levitation height of all objects in the suspension, but only has a small effect on the separation between uninfected RBCs and ring-stage infected RBCs. We observed that increased magnetic susceptibility causes cells and particles to arrive at their equilibrium height faster, but it reduces the overall spread, or variance, of the levitation height distribution, effectively "compressing" the levitation band and reducing separation between distinct entities, thus suggesting the use of a lower levitation height (but with enough paramagnetic concentration to cause visible levitation above the bottom of the channel). We show in Fig. 2e(ii) that increasing the medium density improves separation between uninfected RBCs and ring-stage infected RBCs. We predicted that in a higher-density medium, cells would levitate in the upper half of the channel due to higher buoyancy, and would consequently be subject to a downward magnetic force from the top magnet in the device rather than an upward force from the bottom magnet, in the device's default configuration. This would cause infected cells (with higher magnetic susceptibility) to levitate higher than uninfected cells, adding to the directional effect from density differences and thus improving separation. Overall, the model suggested that medium density and magnetic susceptibility values close to those of the cells will optimize separation resolution.

Simulations of an infected cell population in levitation can be seen in Fig. 2f, including a parameter for the variability present in natural biological samples. Figure 2f(i) shows levitation conducted in a low-density medium (phosphate-buffered solution (PBS)). Figure 2f(ii) shows a simulation of levitation conducted in a high-density medium (PBS with Percoll-colloidal silica particles), producing greater separation in mean height between uninfected and infected RBC populations. The model assumes 5% parasitemia (proportion of RBCs that contain parasites) which approximates our cultured samples. This simulation also shows how much parasitemia could potentially be enriched within the upper portion of the cell height distribution: twofold enrichment in the low-density medium and 2000-fold in the high-density medium case. We changed the dilution parameter of the cell suspension (in Fig. S3) to estimate its effect on the height distribution—this would primarily affect separation resolution during imaging, since the average height of cells would remain the same regardless of dilution. Figure S3a, b(i) show a less dilute blood sample, which benefits from having more parasites in the levitation chamber, but individual cells are less resolvable due to overlap in the image. Figure S3a, b(iii), (iv) shows highly diluted blood samples in which individual cells are more easily resolvable. However, there are fewer parasites in the chamber, which compromises sensitivity at lower parasitemia by reducing the total number of infected cells sampled. These simulations and our observations helped determine the optimal dilution for both sensitivity and imaging resolution, which is between these dilutions, as shown in Fig. S3a, b(ii).

This model assumes that cells are at equilibrium and when forces are balanced, there is little motion in the system other than nanometer-scale Brownian motion, thus ignoring the effect of cell volume and fluid viscosity. Further, the results of this model are highly dependent on the input variables, which are based on the agreement between values of biophysical measurements found throughout the literature, but since measurement methods vary across studies, some variability is expected.

**Levitation of blood samples and separation of infected cells.** After modifying individual properties of RBCs independently, then modeling the interaction effects of modifying multiple properties together on RBC levitation, we proceeded to test RBC populations containing infected cells—i.e., cells that inherently have both density and magnetic susceptibility modified due to parasitic activity. Figure 3 shows typical images of RBCs in a levitation device, with acridine orange staining and fluorescent imaging marking the locations of parasites (which contain nucleic acids, unlike uninfected RBCs which lack nuclei by default) in infected blood. Since these cultures are depleted of leukocytes, the only cells that will stain for nucleic acids are parasite-containing, infected RBCs. Figure 3a(i) shows a representative image of an uninfected RBC sample, and Fig. 3a(ii) shows a representative image of an infected RBC sample (synchronized to ring-stage only). The infected sample again has acridine orange staining to show the locations of parasitized cells. Figure 3b shows a magnified image of an infected population, with PicoGreen-stained infected RBCs highlighted with arrows. These largely match the results of our theoretical model shown in Fig. 2; while infected RBCs occur throughout the infected blood sample, their average height is higher than that of uninfected ones. Analysis of infected blood samples' levitation heights and corresponding fluorescence intensity in Fig. 3c shows that infected RBCs (fluorescent) have a greater height (mean 304 μm) than uninfected RBCs (non-fluorescent) (mean 269 μm).

Since any blood cell population infected with *P. falciparum* will be a mixture of uninfected RBCs (the majority) and infected RBCs (a minority, typically <5%), we tested the mean height of infected RBCs only and compared this with the mean height of uninfected RBCs only. To purify infected cultures into a population of only parasitized RBCs, we performed streptolysin O (SLO) treatment of infected cultures to selectively lyse uninfected RBCs and leave infected ones intact. SLO is a bacterial pore-forming toxin from *Streptococcus* that targets cholesterol in erythrocyte membranes. SLO preferentially lyses uninfected RBCs because they have more membrane cholesterol than infected RBCs, in which membrane cholesterol is depleted due to parasitic metabolism[58,59]. Lysis of uninfected RBCs and concentration of infected RBCs in SLO-treated samples was confirmed with acridine orange staining as shown in Fig. 3a (and described in the "Methods" section). Further details of SLO-based purification of infected cells can be found in Fig. S4.

As Fig. 3d, e show, SLO-purified ring-stage-infected RBCs have a higher mean height than uninfected RBCs in both low-density (Fig. 3d) and high-density (Fig. 3e) media. We also see that the mean heights of mature-stage infected RBCs (i.e., trophozoite and schizonts) are higher than that of ring-stage infected RBCs, confirming that this effect remains consistent into later-stage infection. While the difference was not statistically significant in the low-density case, ring-stage separation performed significantly better in the high-density protocol ($p = 0.00074$, with 13 independent samples of uninfected RBCs, 8 of ring-stage synchronized infected RBCs, and 9 of mature-stage infected RBCs) than in the low-density levitation protocol ($p = 0.44$, with 14 independent samples of uninfected RBCs, 11 of ring-stage synchronized infected RBCs, and 3 of mature-stage infected RBCs). We hypothesize that the separation between uninfected

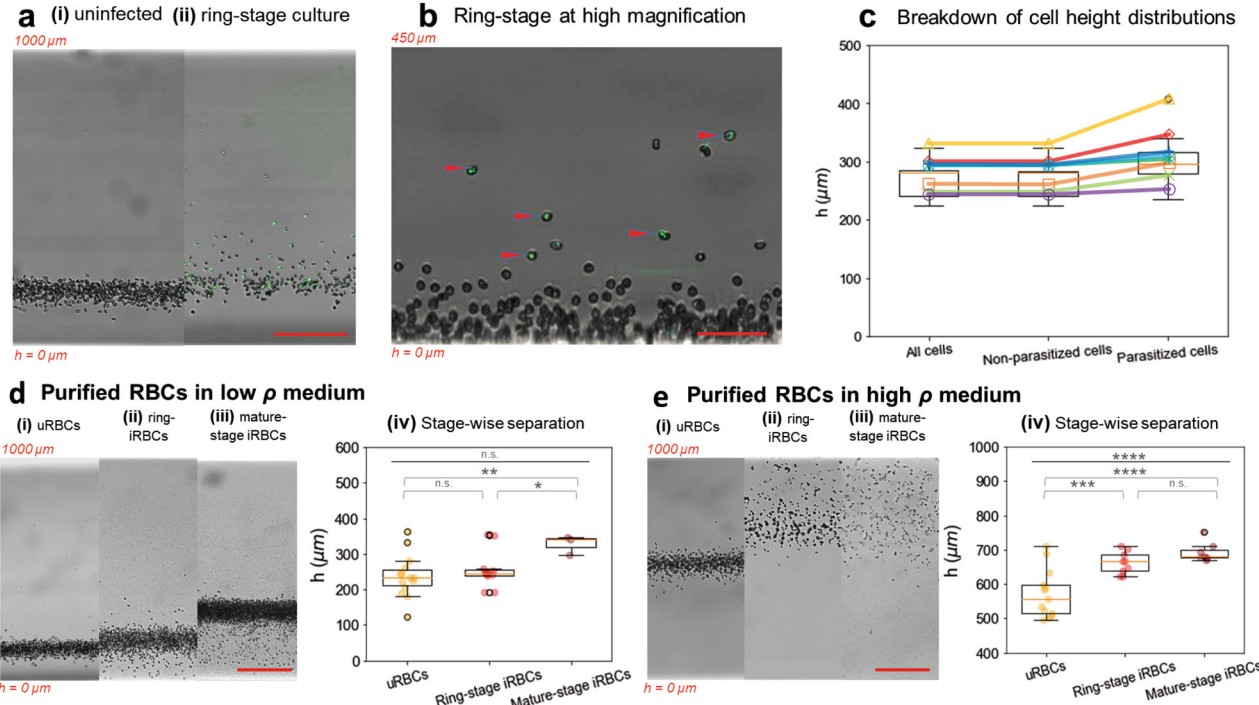

**Fig. 3 Levitation of *P. falciparum*-infected human erythrocytes from 3D7 cultures.** *h* refers to levitation height (µm), *X* refers to magnetic susceptibility (unitless), and *ρ* refers to density (g/mL). RBC refers to red blood cells, or erythrocytes, uRBC to uninfected RBCs, ring-iRBC to ring-stage synchronized infected RBCs, and troph-iRBC to trophozoite-stage infected RBCs. **a** Levitation images of (i) uninfected RBCs and (ii) infected culture containing 5% parasitized RBCs in bright-field merged with fluorescence (acridine orange staining marking parasites by nucleic acid content, in green). Scale bar = 200 µm. **b** Magnified image of infected culture in levitation with PicoGreen stained parasites (red arrows). Scale bar = 100 µm. **c** Fluorescence-based quantification of parasitized components of ring-stage synchronized infected cultures. The mean height of parasitized RBCs (determined by fluorescent labeling with acridine orange), 304 µm, is distinct from that of non-parasitized RBCs (brightfield-only cells without fluorescent signal) 269 µm. Each of the eight series of identically colored circles represents subsets of the same, biologically independent, sample. **d** Levitation images of (i) uninfected RBCs, (ii) ring-stage parasitized RBCs (purified with streptolysin O; SLO), and (iii) mature-stage (trophozoite to schizont) parasitized RBCs (purified with streptolysin O; SLO) in low-*X*, low-*ρ* medium. Scale bar = 200 µm. (iv) Stage-specific measurements of levitation height comparing uninfected RBC samples (*n* = 14, yellow circles) and SLO-purified parasitized RBC samples from ring-stage synchronized cultures (*n* = 11, orange circles) or SLO-purified mature-stage cultures (*n* = 3, brown circles) in low-*X*, low-*ρ* medium. Mature-stage cells levitate higher than uninfected RBCs (*p* = 0.0050), and ring-stage cells (*p* = 0.016). Other differences were not found to be significant. **e** Levitation images of (i) uninfected RBCs, (ii) ring-stage parasitized RBCs (SLO-purified), and (iii) mature-stage parasitized RBCs (SLO-purified) in low-*X*, high-*ρ* medium. Scale bar = 200 µm. (iv) Stage-specific measurements of levitation height comparing uninfected RBC samples (*n* = 13, yellow circles) and SLO-purified parasitized RBC samples from ring-stage synchronized cultures (*n* = 8, orange circles) or mature-stage cultures (*n* = 9, brown circles) in low-*X*, high-*ρ* medium. Ring-stage infected RBCs levitate higher than uninfected RBCs (*p* = 0.00074), and mature-stage RBCs levitate higher than uninfected RBCs (*p* = 0.000038). For both **d**(iii) and **e**(iii), ANOVA was used to compare across groups (asterisks indicate statistical significance across groups). Welch's *t*-test was also used to compare between groups. *\**p* < 0.05; \*\**p* < 0.01; \*\*\**p* < 0.001; \*\*\*\**p* < 0.0001. Box plots represent upper and lower quartiles, with the center line marking the median, and the whiskers showing the range of the data. All samples shown are biologically independent unless otherwise described.

RBCs and ring-stage infected RBCs is improved in high-density medium (as in Fig. 3e) over low-density medium (as in Fig. 3d) because in the high-density case, separation takes advantage of cumulative density and magnetic susceptibility differences.

Thus, we present the first demonstration of quantifying ring-stage infected erythrocytes' biophysical differences using a combination of density and magnetic susceptibility to increase overall separation. This is additionally valuable in achieving separation and detection within a platform that can be adapted for non-laboratory settings.

**Quantification of cell properties and classification of infection state.** To demonstrate that this magnetic levitation platform can be used to detect *P. falciparum* infection in blood samples without the additional steps and resource demands of fluorescent staining, we used brightfield-only images to develop a classification algorithm based on the statistical metrics extracted from the cells' height distributions. The four statistical moments—mean, variance, skewness, kurtosis—are described with Eq. 2–5 in the "Methods" section. We quantified these metrics for each height distribution of the RBC populations and applied a linear discriminant analysis (LDA) feature reduction algorithm to score them as uninfected or infected. As suggested by our model and depicted in Fig. 3, an infected blood sample contains a proportion of infected RBCs that are predicted to levitate higher on average than uninfected cells. We thus hypothesized that distributions containing infected cells would have a higher mean and larger standard deviation and variance, due to increased variation in cell phenotypes and consequently in levitation heights. Further, we expected that infected RBC-containing distributions would have higher skewness due to infected RBCs shifting the height distribution upwards, and higher kurtosis ("tailedness" of the distribution) due to infected RBC heights primarily occurring in the distribution's extremes or tails.

We summarized our findings on the statistical features of the height distributions of uninfected and infected blood samples in

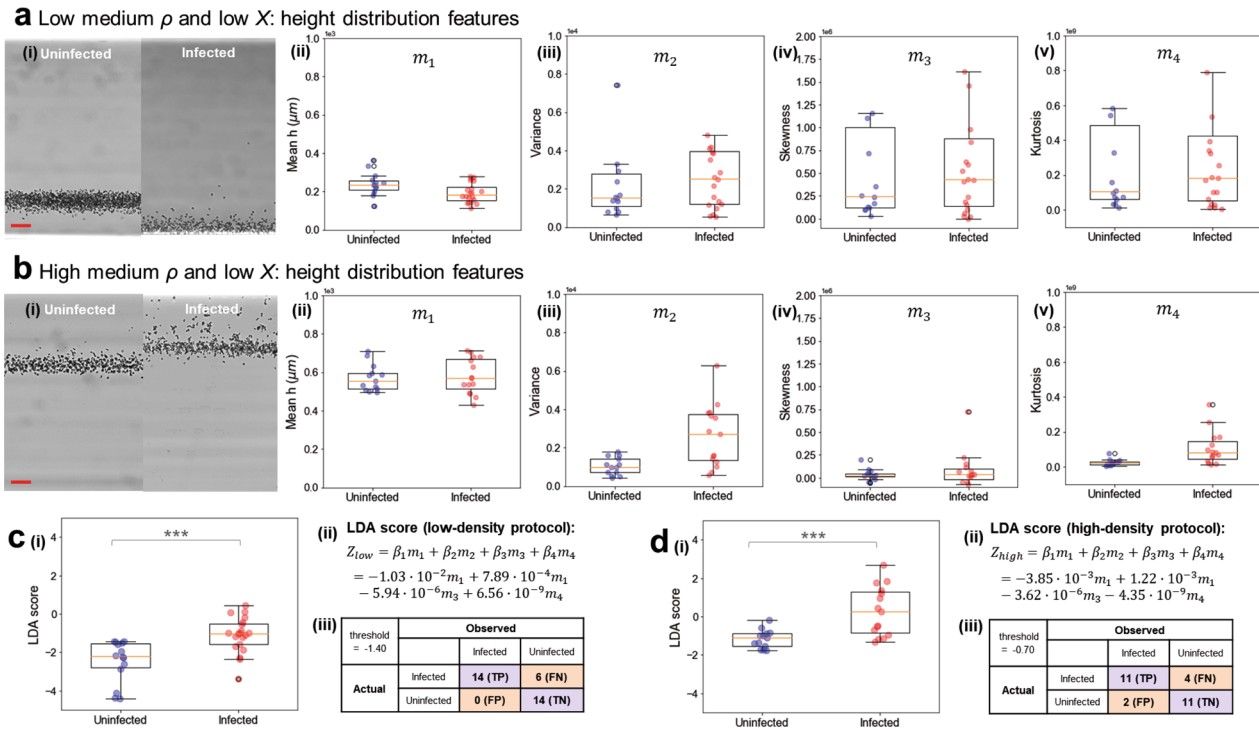

**Fig. 4 Optimization of levitation medium conditions. a** Uninfected blood cells and *P. falciparum*-infected blood cells (3D7 culture line, 5% parasitemia), levitated in isotonic low-density medium with optimized (low) magnetic susceptibility. (i) Example images of cells in levitation. Statistical parameters of the levitation height distribution for uninfected ($n = 14$) and infected ($n = 20$) samples: (ii) mean ($m_1$) ($\bullet 10^3$), (iii) variance ($m_2$) ($\bullet 10^4$), (iv) skewness ($m_3$) ($\bullet 10^6$), and (v) kurtosis ($m_4$) ($\bullet 10^9$). **b** Uninfected blood cells and *P. falciparum*-infected blood cells (3D7 culture line, ring-stage synchronized, 5% parasitemia), levitated in medium with optimized (low) magnetic susceptibility and optimized (high) density. (i) Example images of cells in levitation. Statistical parameters of the levitation height distribution for uninfected ($n = 13$) and infected ($n = 15$) samples: (ii) mean ($m_1$) ($\bullet 10^3$), (iii) variance ($m_2$) ($\bullet 10^4$), (iv) skewness ($m_3$) ($\bullet 10^6$), and (v) kurtosis ($m_4$) ($\bullet 10^9$). **c** (i) Linear discriminant analysis (LDA) scoring ($Z_{low}$) was performed on the statistical metrics (mean, variance, skewness, and kurtosis) for the samples levitated in a low-density medium, to classify them into healthy or infected groups. Box-and-whisker plots show the distribution of LDA scores for each sample tested. The scores for the two groups were statistically significant according to Welch's t-test, with a p value of $8.5 \times 10^{-4}$. (ii) The resulting formula to calculate the LDA score as a recombination of the four metrics. (iii) A confusion matrix classifying each sample by its score, by applying a threshold of optimal separation ($-1.40$). **d** (i) Linear discriminant analysis (LDA) scoring ($Z_{high}$) was performed on the statistical metrics (mean, variance, skewness, and kurtosis) for the samples levitated in a high-density medium, to classify them into healthy or infected groups. Box-and-whisker plots show the distribution of LDA scores for each sample tested. The scores for the two groups were statistically significant according to Welch's t-test, with a p value of $7.9 \times 10^{-4}$. (ii) The resulting formula to calculate the LDA score as a recombination of the four metrics. (iii) A confusion matrix classifying each sample by its score, by applying a threshold of optimal separation ($-0.70$). TP refers to true positive, FP to false positive, FN to false negative, and TN to true negative values. Box plots represent upper and lower quartiles, with the center line marking the median, and the whiskers showing the range of the data. (Scale bar 50 μm).

Fig. 4. Variance, skewness, and kurtosis were indeed found to be higher on average in infected samples than in uninfected samples, suggesting that infected cells shift the distribution. An unexpected finding was that the mean height of infected samples in the low-density medium condition only was lower in some cases than with uninfected samples (Fig. 4a(ii)). In these cases, the entire band was found to be shifted downwards towards the bottom of the device, although the distribution still had upward skewness, as can be seen in Fig. 4a. We explore our interpretation of this in the Discussion.

Other than the mean height in low-density levitated samples, all other statistical metrics matched our hypothesis, including the mean height of high-density levitated samples (Fig. 4b(ii)). Figure 4 compares the performance between low-density and high-density levitation protocols, which we used to determine the protocol to use for optimal separation for classification. These four metrics were employed in our LDA algorithm, which performs dimensionality reduction for a simplified classification problem, mapping the multiparametric sample data onto one scoring axis. LDA was applied to each of the datasets (Figure 4c, d

respectively) to perform a linear recombination of the four features into a single score for each sample. The distributions of these scores are reported in Fig. 4c(i) for low-density medium samples and in Fig. 4d(i) for high-density medium samples. The formula for producing the score from the four metrics, and the coefficients for each one in the linear recombination, are shown in Fig. 4c(ii), d(ii), respectively. A threshold was selected for maximal separation between the uninfected and infected classes, and used to classify the samples by their LDA scores, in each case. The confusion matrices for these classification exercises each are in Fig. 4c(iii), d(iii), respectively. Although larger datasets should continue to be tested to validate this classification accuracy, and this tool is not intended to be a diagnostic, the significant separation between the two classes shows that biophysical differences can be measured and quantified for classification. Multiparametric biophysical separation can be condensed into a single metric and used to distinguish ring-stage infected cultures from uninfected blood samples by the characteristics of their height distributions, even when the cultures are mixed populations of majority uninfected cells and minority infected cells.

## Discussion

Distinct but minute changes occur in cells that undergo functional challenges such as infection, which inform our understanding of a cell's physiological response. Pathologies can cause multiple biophysical changes in a cell that may be related but are measured independently because most tools cannot conduct simultaneous measurements of different properties. The sensitive and high-throughput measurement of these changes, as presented here, can support analytics for both research and clinical applications. For example, improved measurement of ring-stage infected cells could enable a more complete and nuanced understanding of malaria in research and clinical settings, including studying infection susceptibility factors, isolating gametocytes (the stage responsible for transmission) with flow-based extraction[22], and profiling patient-specific drug responses.

While there are both standard (Giema-stained smear microscopy) and novel imaging approaches (e.g. digital holography[60]) for the detection of ring-stage RBCs, their physical separation has been challenging. In laboratory settings, current techniques for separating late-stage infected RBC from uninfected RBCs using parasite-induced physical changes in the host cells have employed the cells' decreased density, increased magnetic susceptibility, or decreased deformability. However, these approaches typically cannot separate ring-stage parasites from uninfected RBCs as successfully as mature-stage parasites. Few techniques can detect such subtle differences in cell density; Percoll gradients are commonly used where centrifuges are available, but cannot distinguish ring-stage parasites from uninfected cells[57]. A microcantilever-based mass density sensor, although highly sensitive, did not demonstrate ring-stage separation, and appears unsuitable for field settings due to needing complex peripheral equipment[24]. In addition, reported techniques to distinguish infected RBCs by magnetic separation (e.g., magnetic columns) have been unable to sensitively capture ring-stage parasites, which only contain trace amounts of the paramagnetic hemozoin[61]. The many structural changes caused by parasitic activity are thus not well-quantified at the ring-stage due to inadequate sensitivity in measurement tools.

Building on previous work using magnetic levitation for separation of different cell types by density alone (from tumor cells to bacterial and fungal samples)[25], this study focuses on minute differences in erythrocytes in response to P. falciparum invasion. While magnetic levitation has previously been used primarily for density measurement, this work demonstrates separation of cells based not only on density differences but also magnetic susceptibility differences. Further, the cumulative application of density and magnetic susceptibility differences uniquely enabled the separation of ring-stage infected cells which have traditionally been difficult to separate with only one biophysical metric. Since fluorescent labeling can be easily added when available, and downstream molecular analyses are still possible on the intact cells, this proposes a new avenue of charting cells with more dimensions than previously possible.

We aimed to shed more light on the biophysical transformations occurring in P. falciparum-infected cells, a focus of active research over the decades, and the cause of the devastating symptoms of malaria. One of our main findings was that ring-stage cells levitated at a distinct height greater than uninfected RBCs and lower than mature-stage infected RBCs. We expected to find that the mature-stage infected RBCs levitate higher than uninfected RBCs because other techniques have already demonstrated biophysical separation based either on the relatively large density or magnetic susceptibility changes in these stages, putting the results presented in Fig. 3d, e in line with the existing literature. However, in addition to this, the separation of ring-stage cells matched our hypothesis and is a unique finding to our

knowledge. As hypothesized, ring-stage separation performed better in high-density medium than in low-density medium. We hypothesize that this is because the high-density protocol takes advantage of both density and magnetic susceptibility differences in the cell, while the low-density protocol primarily depends on only the cell's density differences to induce separation. This outcome suggests that infection changes the density and magnetic susceptibility of host RBCs not only in the mature stages of infection as previous literature has supported, but starting from the ring-stage itself, which has not been shown before due to the inability to sensitively separate ring-stage infected cells on a biophysical basis.

This system's value is enhanced by its operability in comparable or lower-resourced settings than common biophysical separation methods, such as Percoll gradient separation (which requires centrifugation) and magnetic columns (which can be expensive) that are the standard methods in many laboratory applications from basic to clinical science. It also has strong potential, with further development, to be translated to malaria-endemic regions that are typically limited in laboratory infrastructure. The levitation device itself is highly compact ($5 \times 1 \times 1$ cm) and portable (<100 g), and the mechanism of cellular separation is inherently power-free. While we present data that include fluorescence staining to confirm parasitic presence and locations, our algorithm for distinguishing infected cells uses only brightfield distributions, thus not relying on expensive fluorescence microscopes for classification. All reagents and components used for detection are temperature-stable and inexpensive at the low volumes used, and the protocol involves minimal preparation. The test duration primarily spans the time for cells to equilibrate to their final height; 15–20 min under tested conditions (Fig. S5). With further development, the protocol can be adapted to eliminate or automate user-intensive steps so that it can be performed without technical expertise. This technique is appropriate for a range of operative settings to be more accessible in malaria-endemic regions, and enables specific manipulation of the ring-stage. Thus, we hope to enable broader analytical capabilities for scientific experiments and clinical investigations in malaria by overcoming some of these technical limitations.

We observed that some infected blood samples behaved unexpectedly in levitation in a low-density medium; in these cases, the whole population levitated substantially lower than predicted. Since this affected not only the parasitized cells but also the non-parasitized cells within the cultured population, it suggests that external conditions, such as culture conditions, or an indirect effect of infection, is affecting all cells in the cultured sample. Since this occurs in a low-density medium, the lower levitation height of these samples could be due to the whole cell population having a higher density, or a higher magnetic susceptibility (having observed the effects of these changes on levitation in Fig. 2b, c). Indirectly, it could also be caused by higher membrane permeability, since an increased uptake of paramagnetic ions can cause a similar shift. This observation requires further investigation to explore possible mechanisms; these could include various sources of stress or accelerated ageing from culture conditions, or extracellular effects from the parasites.

Sampling a typical population of RBCs would likely represent an age distribution spanning their 100–120 days in circulation, with age correlating positively with density, and density fractionation commonly used to separate RBCs by age[62]. Some evidence in the literature has pointed to RBCs becoming denser as a consequence of accelerated senescence. This was found to occur when infection was correlated with density and several parameters of cellular aging in both parasitized and non-parasitized RBCs in infected cultures[63]. RBCs in infected culture may have different levitation properties due to accelerated aging from

culture conditions in addition to infection effects, which might explain the lower overall levitation band height seen in some samples. There is also evidence of aging-associated methemoglobin generation in RBCs that could affect magnetic susceptibility[64]. We also observed that exposing uninfected RBCs to culture conditions (without any parasites) can have an effect on their levitation heights too (Fig. S6). RBC density is known to increase with cellular age, and the stress of infection may accelerate this process in all blood cells, potentially affecting uninfected RBCs through extracellular mechanisms, such as extracellular vesicles associated with malaria infection[65]. However, these aged cells typically stay within a normally distributed height range and do not tend to demonstrate the high upward skewness that we observed in infected culture samples. As can be seen in Fig. 3a and 4a, b, the levitation profile of a malaria-infected sample is distinct in the shape of its distribution, not just its mean height: it contains more cells that go outside the regular RBC levitation band (i.e., with a different variance, skewness, and kurtosis). Figure 3c further suggests that the parasitized RBCs within a cultured sample tend to levitate higher on average than their non-parasitized counterparts (regardless of the overall height of the levitation band). Thus, while RBC density and/or magnetic susceptibility could be influenced by aging-related mechanisms that may be further accelerated by infection, the overall data suggest that levitation height changes in infected culture samples are not purely due to culture conditions but also from the infection itself.

We note that our study has limitations; a label-free marker such as levitation height could reflect biophysical changes from multiple possible causes, rendering it difficult to pinpoint the specific cause of any observed variance without additional labeling. This may also be responsible for the false-negative values reported. However, this can be mitigated by using fluorescence staining to add layers of information with labeling, and to improve the specificity and accuracy of this multiparametric sample analysis method. Another limitation is the variability in results that may arise from manual steps in the image analysis process. While the process is automated as much as possible, setting image crop parameters to the levitation channel boundary is performed semi-manually, and in rare cases, imaging artifacts must be manually removed. These could lead to different results if processed by different individuals.

The magnetic levitation system we present here is, to our knowledge, the first example of label-free and noninvasive biophysical separation of different stages of *P. falciparum*-infected erythrocytes that includes ring stages. This platform can enable more specific and controlled manipulation of infected erythrocyte populations for research, and it presents an opportunity to develop an accessible platform to put sophisticated analysis of malaria patient samples within reach of field-based healthcare workers. Beyond malaria, this can be applied to measure other biophysical parameters of RBCs, as in sickle cell anemia and other erythrocyte disorders[19,50,66], or other cell types with distinctive biophysical properties. In future work, we are investigating the use of this system to enable field implementation of sensitive and specific tools for cell state quantification. With nearly half the world's population at risk for malaria, it is an urgent priority to reduce access barriers to such analytical tools.

## Methods

An expansion of all abbreviations can be found in Supplementary Table 1 in the Supplementary Information.

**Modeling levitation heights as a function of cellular properties**. Theoretical levitation heights of cells and other objects, such as those in Fig. 2e, f, are calculated

according to Equation 1 below, based on prior work by Mirica et al.[47]:

$$h = \left( \frac{d}{2} + \frac{\Delta\rho \cdot g \cdot p \cdot d^2}{\Delta\chi \cdot 4 \cdot B^2} \right) \cdot 10^6 \qquad (1)$$

in which:

$d$ = height of channel (separation between magnets), in m (0.001 in our device)
$g$ = acceleration due to gravity, 9.8 m s$^{-2}$
$p$ = permeability of free space, 0.001257 m g s$^{-2}$ A$^{-2}$
$B$ = surface magnetic field strength of the magnets used, 300 mT, or g s$^{-2}$ A$^{-1}$
$\Delta\rho = \rho_{cell} - \rho_{medium}$, the difference in density between cell and suspension medium, in g mL$^{-1}$
$\Delta\chi = \chi_{cell} - \chi_{medium}$, the difference in magnetic susceptibility between cell and suspension medium (dimensionless, using the SI unit convention; if starting from values in the cgs convention, multiply by $4\pi$ to convert to SI)
$h$ = output, theoretical levitation height of cell, in μm

These equations were built into a function written in python 2.7 with the above variables as inputs. The main variables were density $\rho$ and magnetic susceptibility $\chi$ for both the medium and cells. The main output of the function was the levitation height $h$ as calculated by the relationship between the inputs described by the equations above. $\rho_{medium}$ is calculated with known masses, volumes, and densities of the solvent (water), solutes (various salts and chelated ions), and other added particles, such as the isotonic colloidal silica particles (from Percoll$^{©}$) used to modulate density without changing tonicity. $\chi_{medium}$ is calculated between the known $\chi$ of water and the molar magnetic susceptibility of the chelated gadolinium ion used as the paramagnetic agent (based on the spin and Curie temperature of the molecule). $\rho_{cell}$ and $\chi_{cell}$ values used are reported or estimated from literature, based on various quantitative measurements of the biophysical values of normal-range (uninfected) RBCs and infected RBCs in various stages, as documented in Fig. 2d and further explained in Fig. S1. The simulation plots of cell levitation in Fig. 2f were computed with the predicted heights as described above, and a normal distribution was generated using Python's numpy module, with cell population size taken from typical cell dilutions and the variance in levitation heights derived from the typical range of red blood cell densities. Further details can be found below in "Detailed methods: explanation of custom code and algorithms: part A".

**Device assembly**. Cells were suspended within a 1 mm tall channel between two neodymium permanent magnets with like poles facing each other, creating a magnetic field within. The basic levitation device is assembled with two neodymium bar magnets (5 cm × 0.5 cm × 0.2 cm), each with a surface magnetic field strength of 300 mT, purchased from K&J Magnetics, Inc (part number 4AGM00). The magnets were arranged with like poles facing each other (in anti-Helmholtz configuration) with a gap of 1 mm between them, as shown in Fig. 1a. The magnets are held together in this configuration with two pieces of polymethyl methacrylate plastic laser-cut on a VersaLaser$^{TM}$, with slots for angled mirrors for imaging in a vertical light path, as in typical microscopes (e.g., Fig. S9). The mirrors used were reflective aluminum-coated microscope slides purchased from EMF (part number AL134). A square glass capillary (1 mm × 1 mm × 5 cm, with 0.2 mm thick walls), purchased from Vitrocom (Cat. No. 8100), is filled with the cell levitation mixture. This cell suspension-containing capillary is inserted into the gap between the magnets, and the device is placed on a typical microscope stage for imaging.

**General preparation of levitation mixtures for experiments**. Paramagnetic solutions were prepared with non-cytotoxic chelated gadolinium ions that are FDA-approved for injection as a contrast agent for magnetic resonance imaging (at a final concentration of 30 mM to achieve the optimal $\chi_{medium}$), in PBS and with colloidal silica particles (Percoll$^{®}$) for high-density medium experiments. The paramagnetic buffer used is a dilution of 1 mmol/mL gadobutrol. This chelated formulation of paramagnetic ions was selected to minimize effects on cell viability and membrane penetration. The density modulation agent used is an isotonic suspension of Percoll$^{®}$ PVP-coated colloidal silica particles (Sigma-Aldrich P1644-25ML) prepared as a dilution in PBS from Thermo Fisher Scientific (Cat. No. 70011-044 for 10X isotonic concentration, and Cat. No. 10010-49 for 1X isotonic concentration). In some cases, acridine orange staining was used for validation by fluorescent staining of the parasites' nucleic acid content, wherein the cell mixture was allowed to incubate with an acridine orange solution prepared in PBS, at a ratio of 9:1 (cell mixture : acridine orange stock) for a minimum of 3 minutes at room temperature. Acridine orange incubation was found to have no significant effect on levitation height of RBCs nor on nucleated leukocytes (as shown in Fig. S7). The main staining agent used is a 0.02% dilution of acridine orange (Sigma-Aldrich A9231-10ML). We suspended the cells in this solution, as in Fig. 1a, typically prepared in the paramagnetic solution in volumes of 100 μL for ease of calculation and for increased measurement accuracy. The mixture is uniformly mixed by vortexing or pipetting up and down. Subsequently, 30 μL of this mixture is inserted into the glass capillary by pipette. The capillary is then sealed on both ends with Critoseal to contain the liquid. The levitation height pattern is imaged once the cells reach equilibrium—around 15–20 minutes for RBCs under the protocol conditions used for the experiments here (Fig. S5).

**Changing tonicity and magnetic susceptibility of RBCs**. Figure 2b documents the results of experiments in which $\chi_{cell}$ was independently modulated for RBCs by modifying the hemoglobin molecule. NaNO$_2$-treated RBCs were hypothesized to have higher $\chi_{cell}$ due to containing methemoglobin, as opposed to control RBCs with oxyhemoglobin, since methemoglobin is more paramagnetic than oxyhemoglobin[34]. For 5 minutes before levitation preparation, healthy RBCs were treated with either 0.1M NaNO$_2$ solution in PBS or with PBS alone as a negative control (1:9 ratio of RBC suspension : NaNO$_2$ solution). This mixture was then prepared with the paramagnetic solution for levitation as per the general protocol described in the above section, taking the dilution of RBCs into account. NaNO$_2$ solution was purchased from Thomas Scientific (Cat. No. C989U96). NaNO$_2$-treated RBCs were hypothesized to have higher $\chi_{cell}$ due to having methemoglobin, as opposed to control RBCs with oxyhemoglobin, since methemoglobin is more paramagnetic than oxyhemoglobin[33,49]. This was tested in both low-density and high-density mediums to determine whether the levitation height change was consistent with the magnetic gradient and not only due to a density change.

Figure 2c documents the results of experiments in which $\rho_{cell}$ was independently modulated for RBCs by modifying the hydration of the cell by osmosis. Uninfected RBCs (from donors via the Stanford Blood Center) were placed in NaCl solutions of varying concentration, adjusted to have the same density using colloidal particles that do not affect osmotic pressure. RBCs in hypertonic solution (285 mM NaCl) were expected to be dehydrated and therefore denser than those in isotonic solution (150 mM NaCl), and RBCs in hypotonic solution (75 mM NaCl) were expected to be swollen and therefore less dense than those isotonic solution.

***P. falciparum* culture and synchronization**. *P. falciparum* 3D7 is a standard laboratory-adapted strain and was cultured in human peripheral blood erythrocytes in complete RPMI medium with 0.5% albumax and 2% hematocrit (Hct) at 37 °C in 5% carbon dioxide and 1% Oxygen. Ring-stage parasites (within 18 hours of bursting) were synchronized by incubation in 5% sorbitol for 10 minutes at 37 °C. Parasitemia, calculated by counting Field's-stained blood smears (for example, in Fig. S8) using a Miller reticle, was in the range of 1–6%, and typically 4–5%. To ensure consistent dilution, 5 µL of this 2% Hct culture was used in every 100 µL preparation of the levitation mixture, unless indicated otherwise.

**Streptolysin O enrichment of infected RBCs**. Experiments such as those in Fig. 3d, e were performed with infected cultures that were enriched for infected RBCs by lysis of uninfected RBCs. This was achieved by treatment with SLO, a bacterial pore-forming toxin from *Streptococcus pyogenes* that targets cholesterol in cell membranes. SLO treatment preferentially lyses uRBCs, which have a higher membrane cholesterol content than iRBCs (in which some of the cholesterol is recruited by the parasites), effectively enriching cultures for iRBCs. SLO was purchased from Sigma-Aldrich (S5265-25KU) as 25,000 units of lyophilized powder, reconstituted in 1 mL of PBS and kept frozen. To prepare activated SLO (for lysis), 144 µL of PBS with 0.1% fetal bovine serum albumin (from EMD Millipore, Cat. No. TMS-013-B) was mixed with 16 µL of 0.1 M DL-dithiothreitol solution (Sigma-Aldrich 646563-10X.5ML) and 40 µL of the reconstituted SLO once thawed. This mixture is incubated at 37 °C for 2 hours, and is stable at 4 °C for up to 1 month[58].

SLO treatment was performed with blood at typical hematocrit (~50%); infected cultures which are typically at low hematocrit such as 2% were centrifuged and aspirated to concentrate to the typical hematocrit of whole blood. Overall, 2 µL of this blood was mixed with 2.8 µL of activated SLO solution and allowed to incubate at room temperature for 6 minutes for lysis to occur. This mixture was then diluted to a final volume of 100 µL and mixed with paramagnetic ions, staining agents, and other additions as per the experiment. Acridine orange staining was used to verify that cultures were enriched, by confirming in microscope imaging that the cell populations contained a significantly higher proportion of fluorescently stained cells.

**Imaging of cell levitation distributions**. After the cells reach their equilibrium height, we imaged them in the device using a microscope, using mirrors to reflect the light to conduct lateral imaging within a standard vertical imaging setup, i.e., to reflect the vertical beam of laser light from the source at 90° to pass horizontally through the cell-containing capillary and then to reflect off another mirror at 90° back into the vertical light path to the sensor (Fig. S9). Zeiss ZEN (Blue edition) software was used to collect microscope image data via an attached camera. We performed image analysis using a custom Python-OpenCV algorithm to distinguish the different levitation height distributions within a population of cells. This custom algorithm was written to process these images of cell levitation patterns, by segmenting cell-containing areas and determining their vertical height distribution. The shape of this distribution was quantified using a number of statistical metrics; these metrics were combined using a custom LDA algorithm to produce a single metric that was used to distinguish infected populations from uninfected populations. The performance of this classifier is quantified with ROC curves in Fig. 4c. The image analysis workflow, with pre-processing, segmentation, cell identification, and statistical analysis, is broadly described in Fig. S10, with further details under

"Detailed methods: part A" below, as well as in "Quantification of cell properties and classification of infection state" in the "Results" section.

**Image analysis**. All image analysis was carried out in Python 2.0 using modules including numpy, scipy, OpenCV for Python, and sklearn. Images are imported as RGB (red green blue format) images from the source TIF (tagged image file format) file and converted to grayscale. Images are then corrected for rotational angle and cropped to consistent dimensions, with the top and bottom of the image corresponding to the upper and lower limits of the channel. Images are resized as necessary to adjust for different microscope source-dependent image dimensions. Next, Gaussian smoothing is applied to remove high-frequency noise from the image, followed by a Laplacian operation to filter low-frequency noise. The image is then binarized using Otsu's thresholding, resulting in an image that identifies cell-positive pixels only. Pixel values are summed across rows to produce a list of heights for all cell-positive pixels, and the array is reversed such that 0 corresponds to the bottom of the channel and 1000 corresponds to the top. The result is a distribution of all cell-containing pixels over height. Since cell morphology is heterogeneous and there is a large degree of overlap between cell boundaries, we considered this to be more accurate than estimating individual cell boundaries which would have resulted in overestimation and underestimation of cell locations by height[67,68]. Further details can be found below in "Detailed methods: explanation of custom code and algorithms: part B".

**Statistical analysis of cell height distributions for classification**. The statistical parameters of this height distribution, its four statistical moments, are calculated as follows:

$$\text{The first moment, mean is } \underline{x}(x_i \ldots x_N) = \frac{1}{N}\sum_{j=1}^{N} x_j \quad (1)$$

$$\text{The second moment is variance}(x_i \ldots x_N) = \frac{1}{N-1}\sum_{i=1}^{n}(x_j - \underline{x})^2 \quad (2)$$

$$\text{The third moment is skewness}(x \ldots x_n) = \frac{1}{N}\sum_{i=1}^{n}\left(\frac{x_j - \underline{x}}{\sigma}\right)^3 \quad (3)$$

$$\text{The fourth moment is kurtosis}(x_i \ldots x_N) = \left\{\frac{1}{N}\sum_{i=1}^{n}\left(\frac{x_j - \underline{x}}{\sigma}\right)^4\right\} - 3 \quad (4)$$

These four statistical moments are used as the metrics for quantifying and classifying the cell populations in levitation. We used LDA[69–71], to condense these four metrics into one, to produce a score to classify the samples as either uninfected or infected, as shown in Fig. 4c (classification threshold for optimal separation: −1.40) and Fig. 4d (classification threshold for optimal separation: −0.70). We used the "LinearDiscriminantAnalysis" function under python's sklearn module, set to a single component in order to condense the four metrics into a single score, the singular value decomposition (SVD) solver, no shrinkage, and a tolerance of 0.0001. We used LDA as a descriptive method to examine the characteristics of our dataset and whether the difference between classes was significant, and found that to be the case with both low-density and high-density protocols. We considered various approaches towards achieving our goal, which was to classify the samples into groups on the basis of multiple features. We chose LDA as a straightforward and commonly used statistical tool for dimensionality reduction, because it includes feature scaling, and most importantly because it is well-suited for performing classification. Principal component analysis is similar in many ways and often compared to LDA; it is an unsupervised learning technique and aims to maximize variance between data points along the projected axes, and is excellent for dimensionality reduction. LDA, on the other hand, is a supervised learning technique, and while it is also effective for dimensionality reduction, it aims to maximize class separation (by maximizing variation between classes while minimizing intraclass variance), and is therefore excellent for the classification problem at hand. In summary, LDA not only includes feature scaling, but also uses class labels (i.e., supervised learning, with known values of 'uninfected' and 'infected' used in training) to aid in optimal classification of samples into the given classes via feature reduction.

We also conducted a preliminary investigation of the accuracy of this approach for classification, by initially splitting our dataset randomly into a training group (70% of the data) and a test group (30% of the data), doing this separately for each of the two protocols. This has the caveat that the dataset size is too small for realistically validating classification accuracy or being able to extrapolate more broadly without further testing on a much larger dataset, or to reliably compare training and test groups since a single sample can introduce a lot of variation into the group. However, as an exercise, we trained the LDA on the training set, determined the threshold for optimal classification, and then applied this scoring formula and threshold to the unseen test set, and compared the classification performance on both of these groups. The classifier was compared to the ground truth values to calculate confusion matrices. The results on these training and test splits, for the low-density case and high-density case respectively, can be seen in

Fig. S11a, b. As expected with such a small dataset, the classification performance is not very strong, but is promising for continued exploration with a larger dataset.

**Detailed methods: explanation of custom code and algorithms (full code uploaded to figshare[72]).**
**(A) Detailed description of levitation height predictions and simulations in Python**
All theoretical values were computed, and simulation images generated, using open-source Python 2.7 in the form of a Jupyter Notebook for visual accessibility. The modules numpy and matplotlib were used for numerical computation and for plot generation, respectively. The referenced equations can be found in the Methods section under "Simulations and calculation of predicted levitation heights". Following is the step-by-step workflow:

(1) Import modules and functions.
(2) Define medium variables (density, magnetic susceptibility).
(3) Define universal constants (e.g., acceleration of gravity) and device constants (e.g., height of cell levitation chamber).
(4) Define cell variables (density, magnetic susceptibility) for different cell subtypes based on known and estimated values from literature.
(5) Ensure all units are converted to compatible SI units (International System of Units).
(6) Input variables and constants into equations and show results (select cell type here to generate values specific to that cell type or stage).
(7) Calculate theoretical separation between cell types under those specific levitation conditions as a difference between theoretical levitation heights.
(8) Plot theoretical predicted values for different cells' levitation heights and their separation over a range of levitation conditions (varying medium density, and then varying medium magnetic susceptibility).
(9) Simulation of cell levitation uses levitation height outputs from previous equations:

(a) Define cell population size (number of cells) and cell dilution, a cell spread factor based on known density range for cells, and percentage parasitemia.
(b) Generate Gaussian cell height distributions by seeding the above inputs into a random number generator, and providing the mean of the distribution (projected height for that cell type), standard deviation (cell spread factor), and number of cells to generate. Every run of this will look slightly different due to the inherent randomness but will fit the same overall parameters of the distribution (mean, standard deviation, size of population).
(c) Print a plot with this distribution, representing the different cell types as circles of different colors, overlaying the distributions for different cell types.
(d) Simulate the effect of changing cell dilutions by increasing or decreasing the cell population size.

**(B) Detailed description of image analysis algorithm to calculate levitation height distributions in Python**
All images were imported, pre-processed, and analyzed using open-source Python 2.7 in the form of a Jupyter Notebook for visual accessibility. The following modules were used: os for handling directories, numpy for numerical computation and handling arrays, scipy for scientific computation and statistical analysis, csv for spreadsheet import and export, matplotlib for plot generation, and cv2 and skimage for image processing. Following is the step-by-step workflow:

(1) Import modules and functions.
(2) Define file path to images.
(3) Import image as RGB.
(4) Convert image from RGB format (three-dimensional array) to grey format (two-dimensional array).
(5) Rotate image to parallelize chamber boundaries with image boundaries in order to compensate for errors in microscope alignment.
(6) Set image crop parameters to chamber boundaries.
(7) Stretch or shrink image (while maintaining proportions) to compensate for different sized images from different microscopes, using preset options for parameters.
(8) Save cropped image for future analysis.
(9) Apply Gaussian smoothing to reduce high-frequency noise.
(10) Apply Laplacian filter to reduce low-frequency noise.
(11) Binarize image using Otsu's thresholding so that it now represents cell-containing pixels and non-cell-containing pixels. (Alternative: Use binary thresholding with threshold set to 5 if image contrast is too low).
(12) Crop 0.5% image boundary to remove false cell detection at boundaries.
(13) If imaging artifacts or debris persist through low-frequency noise removal; set the values of those rows to zero manually.
(14) Sum (binary) intensity values across each row to acquire a heightwise list of the sum of cell-containing pixels.
(15) Subtract average background noise.
(16) Center the distribution according to image so that the height distribution accurately reflects the height of the chamber.
(17) Plot the array that represents the height distribution of cell-containing pixels in each image.

(18) Compute the statistical moments of the distribution using built-in functions, to be used in LDA.

The equivalent of the method described above is also performed for images with both brightfield and fluorescent channel counterparts. The cell-containing pixels from the fluorescent channel are subtracted from those in the brightfield image to estimate nonfluorescent cell locations and fluorescent cell locations without double counting.

**Statistics and reproducibility**. Welch's $t$-test (two-tailed) in Python's scipy.stats was used to compare sample means for statistical significance, since it does not assume equal variances. This was done when comparing two means (Figs. 2a, b, 4c (i), d(i), S6, S7, and S11a(ii), (iv), b(ii), (iv)). ANOVA was used to compare multiple means (Figs. 2c, 3d(iv), e(iv), and S6). Replicates within an experiment are biologically independent unless indicated otherwise, either from distinct human donors, or from distinct cycles of 3D7 *P. falciparum* culture. Error bars typically represent the standard error of the mean. When box-and-whisker plots are presented, each element represents the following: center line represents the median; box limits represent the upper and lower quartiles; whiskers represent the range of the data; points (circles) represent outliers.

**Reporting summary**. Further information on research design is available in the Nature Research Reporting Summary linked to this article.

## Data availability
The underlying numeric source data for all graphs presented (i.e., levitation height distributions and their features) are tabulated and organized by figure of appearance, and included in a supplementary data file. Further data and information are available from the corresponding author on reasonable request.

## Code availability
The code is uploaded on the open-source repository figshare at the following https://doi.org/10.6084/m9.figshare.14445223.v1[72] under the GNU General Public License Version 3.0, and includes instructions for use and a demo set of example data. Readers may contact the corresponding author for more information on implementation of the code. Key operations and characteristics of the code are also described in detail in the "Methods" section of the paper.

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

## Acknowledgements

We acknowledge the generous support for experimental resources (specifically *P. falciparum* cultures) from B.S., E.E., and other members of E.E.'s research group, as well as members of B.G.'s research group. Uninfected whole blood samples were obtained via the Stanford Blood Center. We also thank the members of the research groups of U.D., E.E., and B.G. among others, for technical and conceptual advice. This work has been generously supported by research funding from the Stanford King Center for Global Development (to S.D.), the Stanford Predictives And Diagnostic Accelerator (SPADA) from the SPECTRUM CTSA Program, and the Thomas and Stacey Siebel Foundation (to S.D.).

## Author contributions

S.D. and U.D. conceived the project and acquired funding with collaborative support from B.G. and E.E. S.D. designed and performed experiments, with training from N.G.D., experimental assistance from B.S. and A.C., and conceptual contributions from all authors. S.D. performed data analysis with interpretation from B.S., B.G., E.E., and U.D. S.D. wrote the manuscript, with assistance from B.G., E.E., and U.D. All authors edited the manuscript.

## Competing interests

U.D. is a founder of and has an equity interest in: (i) DxNow Inc., a company that is developing sperm sorting and microfluidic IVF tools and imaging technologies for point-of-care diagnostic solutions, (ii) Koek Biotech, a company that is developing microfluidic technologies for clinical solutions, (iii) Levitas Inc., a company focusing on developing microfluidic products for sorting rare cells from liquid biopsy in cancer and other diseases, (iv) Hillel Inc., a company bringing microfluidic cell phone testing tools to home settings, and (v) Mercury Biosciences, a company developing vesicle isolation technologies. U.D.'s interests were viewed and managed in accordance with the conflict-of-interest policies. The remaining authors declare no competing interests.
