## [Peer Review File · Communications Biology]

Reviewers' Comments:

Reviewer #1:

Remarks to the Author:

The manuscript "Multiparametric biophysical profiling of cells" by Deshmukh et al. describes a microfluidic setup using magnetic levitation to discriminate cells infected by *Plasmodium falciparum* from non-infected cells.

The topic is of general interest to the readership of *Communications Biology* since detection of the malaria parasite is still time-consuming and requires a trained technician. The method introduced by the authors has a strong translational component and could potentially be used for field works. However, reading the manuscript is sometimes a bit challenging. Especially, more care is required in avoiding redundancies on one hand and providing sufficient information to follow the general ideas on the other.

Before making a recommendation, I would be happy to have the following points addressed:

1. Are the cells flowing in the capillary?
2. On page 3, line 22. What do the authors mean by bending the light?
3. On page 3, the section "Imaging of cell levitation distribution" should be moved to Methods.
4. On page 4 and in Figure 2d (i/ii), how was the calculation of the predicted heights be performed (line11)?
5. What about bystander effects? Do not infected cells in an infected sample behave differently than control? This is discussed a little bit on page 7. Could the authors comment how the relative number of infected cells impact on the levitation height of the overall population? The authors should also check Toepfner et al. *eLife* (2018).
6. On page 6, line 23 the authors refer to the standard acridine orange staining in Fig.2. Where can this be found?
7. On page 6, line 30 and Fig.3D(iii) how many samples of how many cells are included in the statistics? This information should be available in the main text.
8. On page 6, line 32 – 46. This part belongs more to the discussion.
9. On page 7, line 14 – 30. This part belongs more to the discussion.
10. On page 10, line 44, which threshold was applied for the ROC analysis? It is indicated in the Figure but should also be stated in the Methods section.
11. In Figure 1, caption – a) what is meant by reagents that are mixed with blood? Also, what is meant by vertical mirrors?
12. In Figure 2, caption – please introduce all abbreviations. a) What is meant by n=4 samples? Is this the number of replicates? How many cells contains each replicate? c) What is the reference for these numbers? d) How have these calculations been performed? e) How has the simulation been performed?

In summary, the idea and simplicity of the method is very nice and I encourage the authors to revise their manuscript.

Reviewer #2:

Remarks to the Author:

The manuscript submitted by Deshmukh et al. describes an implementation of magnetic levitation technology to detect erythrocytes that are infected with *Plasmodium falciparum*. Malaria diagnosis, especially in resource limited settings is a relevant biomedical concern. Authors describe a novel application of microfluidics based magnetic levitation to delineate differences between normal and infected erythrocytes. For this, they use levitation height, which corresponds to the combination of single cell density as well as cell magnetic susceptibility. Authors conclude that a high-density medium is required for magnetic levitation to detect malaria infection based on single cell density of erythrocytes. However, a confident detection performance based on magnetic susceptibility of cells is less likely.

Results are novel, impactful, and timely, but some specific issues are present before the manuscript is ready to reach a larger audience.

- 1) The title is selected to be too general for the content. The current form is perhaps suitable for a running title, but the main title needs to be more specific.
- 2) To be prudent in describing cell detection and separation is achievable with magnetic levitation, authors should mention and cite live/dead cells (<https://doi.org/10.1101/2020.07.27.223917>) and adipocyte separation (<https://doi.org/10.1039/C8AN02503G>) in the introduction part.
- 3) On a similar note, it is advisable to mention (<https://doi.org/10.1039/C8LC00350E>) for label-free early detection of malaria.
- 4) The content under the heading "Imaging of cell levitation distributions" at the results section should be properly transferred to the method section and/or supplementary information file. A single sentence should be enough in the results section to refer to the methods.
- 5) The argument of how changing the medium density effects the levitation height (Supp. Fig. 5) is expected to be presented early in the results, preferably even before NaNO₂ treatment. Furthermore, reaching an equilibrium height should be a lot longer in denser media compared to the normal media, so Supp.Fig.6 would benefit from the representation of both cases.
- 6) For NaNO₂ treatment, authors state that "In all cases, densities (ρ) remained unchanged". This statement needs clarification and detail. I suppose with the utilization of a high density medium and based on the governing equations of magnetic levitation, the levitation height differences should become more pronounced even for minor differences in single cell density. Is this the reason that authors reach to the no density change conclusion?
- 7) Authors can translate their levitation height measurements into single cell density values and compare these findings to the data presented in Figure 2c. Such a comparison would improve the foundation of the study.
- 8) Statistical significance between groups is denoted with *, **, or ***, but corresponding p values should be specified in the figure legends. This is present for figure 3 but missing in figure 2.
- 9) Based on the other levitation images, Figure 3b lower boundary does not look like 0.
- 10) For Figure 3d and 3e, is there a reason for not including the microscope images of late stage iRBCs?
- 11) For figure 3c, though it is obvious, the difference in the levitation height of parasitized cells should be depicted with a statistical test for completion. Also, perhaps the choice of bar colors can reflect that the first bar is a combination of the last two. And finally, the fraction range of cells can be depicted over the last two bars to help narration.
- 12) In the caption of Figure 3, "For both (D)(iii) and (E)(iii)" should be corrected as "For both (d)(iii) and (e)(iii)" with lower cases. Furthermore, Figure 4.(aa) should be Figure 4.(a)
- 13) Linear discriminant analysis is a neat way of dimensional reduction. However, it is not clear why authors did not try more straightforward approaches such as coefficient of variation or feature scaling with their distributions. Furthermore, authors should mention the selection criteria of LDA compared to other dimensional reduction protocols such as principle component analysis.
- 14) Authors should mention the final concentration of the gadolinium agent used for levitation in the text for reproducibility. Furthermore, a simulation mapping of medium density vs. the gadolinium concentration for the levitation height (or single cell density) difference of parasitic cells may offer insight to improve the sensitivity of the test.

Reviewer #3:

Remarks to the Author:

Deshmukh and colleagues describe a multiparametric approach to profile cells based on minute differences in density and magnetic susceptibility. The malaria parasite *Plasmodium falciparum* is used to demonstrate the feasibility of this approach as infection of red blood cells with this pathogen simultaneously decreases the density of the host cell and increases its magnetic susceptibility. The minute differences between ring-infected and uninfected red blood cells in terms of density and magnetic characteristics have complicated separation by means of a single parameter. The authors show that a multiparametric approach combining these parameters allows a better separation of these stages.

The manuscript is well-written and organized in a logical manner. The findings described are novel and of could be of interest to a broader audience. However, there are some aspects of the paper that merit attention and require clarification.

Major comments

1. One aspect that merits attention is the application of the platform. In the introduction, from lines 48 onwards, the authors propose the use of their newly developed platform in resource-poor settings. They further elaborate on this in the discussion (lines 29-41). My concern here is that the authors have not used any patient blood in the work described here and thus this claim cannot be substantiated. For this it would be useful to show data with patient derived material, or as a proxy, with parasites diluted in whole blood. It appears that figure S8 shows some preliminary data on this, but it is not discussed in the main text.

Furthermore, it is not clear to me what the advantage is of the levitation method in resource constrained communities over the already available simple methods such as the use of giemsa stained blood films/dipstick. While the platform described is of interest as a fundamental biological finding, questions remain as to what the authors see as a realistic application. The authors do some suggestions on this in lines 5 and 6, but how this is done and how this would provide advantages over established methods is unclear.

2. Fig S5 addresses some variables that can result in different levitation patterns. This highlights the importance of using uninfected RBCs that are treated exactly the same as the infected RBCs (so kept in culture for similar periods of time as the infected cells and sorbitol treated). Given that the authors suggest to use the platform in resource poor settings, where it may not be possible to use fluorescence for parasite detection, I strongly suggest that the authors show a side by side comparison of uRBC, RBC that have been treated exactly as the iRBC and the iRBC (preferably using whole blood).

3. Another important potential application is the opportunity to isolate infected young ring stage parasites. It is not clear whether this is already feasible or whether the authors consider this to be possible in the future. For example, in figure 1A a picture is shown of a blood smear after levitation to equilibrium. Does this mean that the parasites can be isolated from the device? Please clarify. On p8 line 5 the authors mention the isolation of specific stages responsible for transmission. Can the authors please be more specific about this? How is this done and what do the authors want to achieve?

In lines 27 (p8) the authors mention that molecular analyses are still possible because the cells are intact and viable. This suggest that the authors have purified the cells and checked their viability. Have the authors done this? If this can be done it is important to show the data on this.

Minor comments

1. Figure 1C: since the density of infected RBCs is slightly lower, it may be good to show a minor difference between uRBC and ring-RBC on the ρ axis (they now look the same).
2. In the results section, p3 starting at line 47: it is stated that pathological conditions may alter the magnetic signature of RBCs. Given that malaria-infection patients often suffer from anemia wouldn't this complicate the levitation read-out of patient derived infected cells?
3. p5, line 3 (Figure 2C): levitation heights are not shown in this figure.
4. p5, line 10. The ref number appears incorrect (possibly this should be ref 35?)
5. Figure S3: while panel c shows a simulation of a 1:10 dilution, there are many more dots (cells) depicted than in the undiluted sample (panel a). Can the authors clarify this? Also, the figure refers to Fig 2F, but this is not present. Please correct.

6. In figures 2 and 3, bar charts are used. It is not noted what the error bars represent. Scatter plots would be preferred because this gives a better sense of the variation between measurements.
7. p6, line6. In the methods section sorbitol synchronization is described, but it is not clear what the age of the rings is. How tight was the synchronization? 'Old' rings may give different results than very 'young' rings.
8. In figure 3 and throughout the manuscript the term 'healthy' is used (for example also on p7 line6). This suggests that the authors have checked the health status of the cells involved. This is confusing; please use uninfected instead.
9. Figure 3: Does the acridine orange have any effect on the levitation?
10. In figure 3 a(i) many more cells appear to be present than in figure 3 a(ii). Please comment. Also, the uninfected cells (not fluorescent) have a different levitation height than in (i); can the authors explain?
11. Figure 3 d and e: it is important to include SLO treated uRBC controls as well, because from figure S4b it appears that uninfected cells in a normal culture differ in levitation from uninfected cells (fluorescence negative) in a SLO treated culture. Or is this a sorbitol effect? In d(ii): are this all parasites or is there also debris?
12. P6 line 32. Mature-stage parasites are mentioned. Please explain how these stages were obtained (how synchronous were they?). It would be good to show a picture (at a higher magnification than that is used in fig S7) of the material that was used for the levitation (a stained slide with stained parasites).
13. P6 line 48. Rather than using 'disease' state I would suggest to use 'infection' state, because disease is not measured here.
14. P7 line 14. The authors summarize different features of infected and uninfected blood samples. Would the authors consider blood group to be a variable? And what about the temperature of the sample? Could this influence the measurement?

We thank all of the reviewers for sharing their time and expertise in reviewing our manuscript. We believe the manuscript has benefited greatly from this review process and hope that you can look favorably upon this revised version, which has taken into account every one of the comments that were provided so thoughtfully. For ease of review, we have included our responses below under every comment, in blue for easy differentiation. In most cases, we have included a copy of the portion of the manuscript that was edited directly here, but in cases where extensive changes were made over larger areas, we have noted the location in the manuscript where the appropriate changes can be found.

-Authors

Referee expertise:

Referee #1: microfluidics, cell separation, biophysics

Referee #2: Mechanobiology, Biofabrication, Microgravity, Magnetic Levitation, Single Cell Studies

Referee #3: Plasmodium, imaging, cell separation

Reviewers' comments:

Reviewer #1 (Remarks to the Author):

The manuscript "Multiparametric biophysical profiling of cells" by Deshmukh et al. describes a microfluidic setup using magnetic levitation to discriminate cells infected by *Plasmodium falciparum* from non-infected cells.

The topic is of general interest to the readership of *Communications Biology* since detection of the malaria parasite is still time-consuming and requires a trained technician. The method introduced by the authors has a strong translational component and could potentially be used for field works. However, reading the manuscript is sometimes a bit challenging. Especially, more care is required in avoiding redundancies on one hand and providing sufficient information to follow the general ideas on the other.

Thank you for these comprehensive comments. We have followed all the detailed comments you thoughtfully provided and hope we have improved the manuscript by making the suggested changes. We have also moved through the manuscript to clarify core concepts informing the ideas we present, and to reduce redundancies along the way.

Before making a recommendation, I would be happy to have the following points addressed:

1. Are the cells flowing in the capillary?

At equilibrium when we image the cell height distribution, the cells are **not** flowing in the capillary. We have addressed this question in the 'Results' section under "Magnetic levitation system and device design" (page 3, line 29 onwards):

"To understand the cells' biophysical profile, it is sufficient to image them in their equilibrium positions to record their final vertical locations (without inducing flow). The information in these

imaged patterns is then extracted as described in “Imaging of cell levitation distributions” in the Methods, to be subsequently analyzed as explained.”

2. On page 3, line 22. What do the authors mean by bending the light?

We have added a more detailed explanation of this in the ‘Methods’ section under “Imaging of cell levitation distributions” on page 14, line 38 onwards.

“After the cells reach their equilibrium height, we image them in the device using a microscope, using mirrors to bend the light to conduct lateral imaging even within a standard vertical imaging setup.”

We have also added another supplementary figure with a diagram that explains this setup (Fig. S13):

Fig. S9. Use of mirrors to angle light for a standard vertical light path microscopy setup.

3. On page 3, the section “Imaging of cell levitation distribution” should be moved to Methods.

We have made this change as recommended. This section is now on page 14, line 37 onwards.

4. On page 4 and in Figure 2d (i/ii), how was the calculation of the predicted heights be performed (line11)?

We have explained this in detail, and now included here that “We used these equations to predict cell levitation heights under varying fluid conditions (density or magnetic susceptibility) as in **Fig. 2e**, further described in the Methods.” for increased clarity. While the full explanation is provided there (page 12, line 2 onwards) and is too long to copy here, in summary, the forces are assumed to be balanced when the cells assume their equilibrium positions, with gravity and buoyancy, and magnetic repulsion/attraction at play. Thus, we can use a set of equations that describe this force balance when these cells are at stillness, in which the biophysical properties of the cells (density, magnetic susceptibility), the biophysical properties of the liquid medium they are suspended in (density, magnetic susceptibility), and the dimensions of the system (e.g. space between magnets) can be used to calculate the predicted vertical position of the cells in the system.

5. What about bystander effects? Do not infected cells in an infected sample behave differently than control? This is discussed a little bit on page 7. Could the authors comment how the relative number of infected cells impact on the levitation height of the overall population?

We go over some of these effects in our discussion, copied here below. These questions touch on an important aspect of this research, which is to try to differentiate how the different subtypes of cells act within a diverse population, what effect they are specifically responding to, and what these various contributions map to in terms of the final levitation height. Thus, there are several relevant points to consider:

- Is the mean height of the overall population affected in the case of infection? How much of this effect is due to the parasitized subset which is a minority of the cells)?
- Are there more global effects on the cell population in an infected case beyond just the parasitized cells themselves?

As we show in Fig. 4A(ii-v) and 4B(ii-v), the infected component impacts the height distribution of the cells (overall population) in a number of ways, some more significant than others. While the effect on mean height behaved in somewhat unexpected ways (which we discuss in the manuscript, and some of this is included below), regardless of the mean height of all cells in a levitated population, the infected component consistently influences the variance in the height distribution, and specifically skews the height distribution in a lopsided fashion, as well as pushing more of the distribution into the tail rather than the center of the spread (kurtosis). As we show in Figs. 3b and 3c with a visual and quantitative breakdown of the heights of individual components of an infected cell population, the parasitized cells have a significantly higher mean height than their non-parasitized components, and this difference between the two subtypes persists even when the overall population varies as a whole. This difference is what we hypothesize as contributing to the unique variance, skewness, and kurtosis profiles of infected populations, beyond just affecting the mean height of the RBC levitation bands, even though parasitized cells typically only constituted roughly 5% of the overall population in our infected samples.

As to whether there may be more global effects on RBCs in the case of our infection cultures, some of the discussion pasted below addresses questions what effect the RBCs are responding to -the infection from the parasite directly or an indirect effect of the culture conditions that are necessary to grow the RBCs with parasitic infection. This may explain some of the unexpected effects we found on the population as a whole (both parasitized and non-parasitized cells within an infection-cultured population); the text below can be found in the Discussion on page 10, line 17 onwards:

“Unexpectedly, we found that infected blood samples as a population—both uninfected RBCs and infected RBCs—levitate substantially lower than predicted in a low-density medium. This suggests that malaria infection may have a global effect on blood cells beyond the direct effect on parasitized RBCs’ biophysical properties. The decreased levitation height of both infected and uninfected RBCs (with infected RBCs still levitating higher than the uninfected ones) suggests either a higher density, higher magnetic susceptibility, or greater membrane permeability to solutes that may increase either of those two variables. There is some evidence in the literature to support the hypothesis that RBCs become denser as a consequence of accelerated senescence. This was found to be the case when infection was correlated with density as well as

several parameters of cellular aging in both infected and uninfected RBCs in infected cultures⁶². RBC density is known to increase with cellular age, and the stress of infection may accelerate this process in all blood cells, potentially affecting the uninfected RBCs through extracellular mechanisms, as there are known to be extracellular vesicles associated with malaria infection⁶³.”

The authors should also check Toepfner et al. eLife (2018).

Based on the recommendation, the above work of literature has been cited in the Introduction now, on page 2, line 23.

6. On page 6, line 23 the authors refer to the standard acridine orange staining in Fig.2. Where can this be found?

We have added more details on the acridine orange staining process and noted this in the main text (on page 7, line 13 onwards) as follows: “Lysis of uninfected RBCs and concentration of infected RBCs in SLO-treated samples was confirmed with the standard acridine orange staining method as shown in **Fig. 3a** (which can be found in the Methods).”:

In the Methods (on page 13, line 15 onwards): “In some cases, acridine orange staining was used for validation by fluorescent staining of the parasites’ nucleic acid content, wherein the cell mixture was allowed to incubate with an acridine orange solution prepared in PBS, at a ratio of 9:1 (cell mixture : acridine orange stock) for a minimum of 3 minutes at room temperature. Acridine orange incubation was found to have no significant effect on levitation height of RBCs (as shown in **Fig. S11**). The main staining agent used is a 0.02% dilution of acridine orange (Sigma Aldrich A9231-10ML).”

7. On page 6, line 30 and Fig.3D(iii) how many samples of how many cells are included in the statistics? This information should be available in the main text.

The text has been updated in this location (now page 7, line 21 onwards) to include the specific number of cell samples used for these experiments:

“As expected, ring-stage separation performed better in the high-density protocol ($p < 0.001$, with 12 samples of uninfected RBCs, 7 of ring-stage synchronized infected RBCs, and 9 of mature-stage infected RBCs) than in the low-density levitation protocol ($p < 0.05$, with 12 samples of uninfected RBCs, 11 of ring-stage synchronized infected RBCs, and 3 of mature-stage infected RBCs).”

8. On page 6, line 32 – 46. This part belongs more to the discussion.

Following this recommendation, the majority of this section has been transferred into the Discussion as per the comment. This can be found on page 10, line 17 onwards.

9. On page 7, line 14 – 30. This part belongs more to the discussion.

Following this recommendation, the majority of this section has been transferred into the Discussion as per the comment. This can be found on page 10, line 30 onwards.

10. On page 10, line 44, which threshold was applied for the ROC analysis? It is indicated in the Figure but should also be stated in the Methods section.

This has now been included in the Methods section as suggested, on page 15, line 35 onwards:

“We used linear discriminant analysis^{69–71}, to condense these four metrics into one, to produce a score to classify the samples as either uninfected or infected, as shown in **Fig. 4a** (threshold of classification calculated as -0.387) and **Fig. 4b** (threshold of classification calculated as 0.338).”

11. In Figure 1, caption – a) what is meant by reagents that are mixed with blood? Also, what is meant by vertical mirrors?

The figure caption has been updated to clarify both of these points within the scope of this text. A more detailed description of all steps involved can be found in the Methods under Device assembly (page 12, line 31 onwards) and under General preparation of levitation mixtures for experiments (page 13, line 5 onwards).

Updated figure caption section:

“Workflow: First, a volume of blood (smaller than that of a fingerprick) is mixed with a paramagnetic solution within a glass capillary and loaded into the magnetic device. Scale bar = 1 cm. Second, cells begin to levitate at different heights, arriving at equilibrium within 10–12 minutes and resulting in height-based separation of different cell types. Third, cell levitation patterns in the device are imaged upon equilibrium using a microscope for analysis.”

12. In Figure 2, caption – please introduce all abbreviations.

Remaining abbreviations are now defined in the caption:

““RBC” refers to red blood cells, or erythrocytes; “uRBC” refers to uninfected RBCs; “ring-iRBC” refers to ring-stage synchronized infected RBCs, and “troph-iRBC” refers to trophozoite-stage (i.e. more mature) infected RBCs.”

We have also added a full list of abbreviations and symbols in Table 1 at the end of the Supplement (page 14) as an accessible reference for the reader.

a) What is meant by n=4 samples? Is this the number of replicates? How many cells contains each replicate?

This has been described with more detail now in the caption, but to explain further: each sample is a small volume taken from aliquots of human donor blood. Therefore, n is the number of replicates taken from unique donors (for uninfected samples) or from unique cycles in the 48-hour infection life cycle in culture (for infected samples). Each replicate is typically prepared with a dilution of the RBC samples that corresponds to roughly 37,500 cells within each 30 μL volume in levitation. This estimate is based on our dilution factor of $2.5 \cdot 10^4$, applied to the typical hematocrit level around $5 \cdot 10^6$ RBCs per μL of human whole blood.

Updated caption section example:

“(a) Uninfected red blood cells in levitation without treatment or with sodium nitrite (NaNO_2) treatment which converts the hemoglobin into the more paramagnetic methemoglobin. The experiment was conducted in low-density medium (n = 4 distinct RBC population samples from donor blood aliquots in the untreated condition, and n = 4 in the treated condition), p = 0.000125 and in high-density medium (n = 4 in the untreated condition, and n = 4 in the treated condition), p = 0.035353.”

c) What is the reference for these numbers?

We have now included the citations for this directly in the text, and Fig S2 contains a fuller breakdown of how these biophysical estimates are calculated from values reported in the literature.

Figure S2. Estimated single cell biophysical parameters of erythrocytes undergoing *Plasmodium falciparum* infection, estimated by integrating various quantitative biophysical measurements and models. Key denotes various cellular components shown in pie charts.

(a) Uninfected RBC. **(b)** Ring-stage RBC. **(c)** Trophozoite-stage RBC.

Masses estimated from spectral measurements, published by Serebrennikova *et al* in *J. Theoretical Biology. Quantitative analysis of morphological alterations in Plasmodium falciparum infected red blood cells through theoretical interpretation of spectral measurements* (2010); volumes estimated by quantitative phase spectroscopy (QPS), published by Rinehart *et al* in *Sci. Rep. Hemoglobin consumption by P. falciparum in individual erythrocytes imaged via quantitative phase spectroscopy*. (2016).

d) How have these calculations been performed?

e) How has the simulation been performed?

Both of the above are explained in detail under the Methods under “Modeling levitation heights as a function of cellular properties” which is too long to reproduce here but can be found on page 12, line 2 onwards.

In summary, the idea and simplicity of the method is very nice and I encourage the authors to revise their manuscript.

Thank you very much for your detailed comments and recommendations that have helped to improve the quality of this manuscript.

Reviewer #2 (Remarks to the Author):

The manuscript submitted by Deshmusk et al. describes an implementation of magnetic levitation technology to detect erythrocytes that are infected with *Plasmodium falciparum*. Malaria diagnosis, especially in resource limited settings is a relevant biomedical concern. Authors describe a novel application of microfluidics based magnetic levitation to delineate differences between normal and infected erythrocytes. For this, they use levitation height, which corresponds to the combination of single cell density as well as cell magnetic susceptibility. Authors conclude that a high-density medium is required for magnetic levitation to detect malaria infection based on single cell density of erythrocytes. However, a confident detection performance based on magnetic susceptibility of cells is less likely.

Results are novel, impactful, and timely, but some specific issues are present before the manuscript is ready to reach a larger audience.

Thank you very much for your detailed comments and recommendations. We believe the manuscript has been significantly improved by making the suggested changes in content, presentation of ideas, format, etc. We do find that using a combination of **both** cell density and cell magnetic susceptibility for improved separation potential is more helpful than using either variable on its own.

1) The title is selected to be too general for the content. The current form is perhaps suitable for a running title, but the main title needs to be more specific.

The title has been revised in response to this feedback as “Multiparametric biophysical profiling of red blood cells in malaria infection”.

2) To be prudent in describing cell detection and separation is achievable with magnetic levitation, authors should mention and cite live/dead cells

(<https://doi.org/10.1101/2020.07.27.223917>) and adipocyte separation

(<https://doi.org/10.1039/C8AN02503G>) in the introduction part.

Based on the recommendation, the above works of literature have been cited in the Introduction now, on page 2, line 14.

3) On a similar note, it is advisable to mention (<https://doi.org/10.1039/C8LC00350E>) for label-free early detection of malaria.

Based on the recommendation, this method has now been cited in the Discussion as a novel imaging-forward approach, on page 9, line 2.

4) The content under the heading "Imaging of cell levitation distributions" at the results section should be properly transferred to the method section and/or supplementary information file. A single sentence should be enough in the results section to refer to the methods.

This section has now been transferred to the Methods. One sentence, on page 3, line 31, refers to this as follows:

“The information in these imaged patterns is then extracted as described in “Imaging of cell levitation distributions” in the Methods, to be subsequently analyzed as explained.”

5) The argument of how changing the medium density effects the levitation height (Supp. Fig. 5) is expected to be presented early in the results, preferably even before NaNO₂ treatment.

We have now included a figure to show how increasing medium density (by use of Percoll) affects levitation height (i.e. increases) for cells of the same type (uninfected RBCs) -without any treatment or infection, as Fig. 2a.

Figure 2. (a) Increasing the density (ρ) of the medium leads to increased levitation heights for the same cell type (uRBC).

Furthermore, reaching an equilibrium height should be a lot longer in denser media compared to the normal media, so Supp.Fig.6 would benefit from the representation of both cases. Thank you for this comment, we have now included this data in Supplementary Figure S6 as well (i.e. both low-density and high-density cases), as reproduced below, so we hope this figure is more representative and informative now. Compared to the low-density case where equilibrium is reached within 12 minutes, we have found that equilibrium height does take slightly longer - within 16 minutes- in denser media (likely due to the increased viscosity).

Figure S6. Time to levitation height equilibrium. **(A)** Images and height distributions of a typical RBC population in levitation over time, since insertion into device (in the case of low-density medium). **(B)** Images and height distributions of a typical RBC population in levitation over time, since insertion into device (in the case of high-density medium). **(C)** Mean height of cells in levitation chamber over time from sample insertion, in both cases: cells start to disperse, begin to sediment, and come to equilibrium within 12 min (low-density case) or 16 min (high-density case).

6) For NaNO₂ treatment, authors state that “In all cases, densities (ρ) remained unchanged”. This statement needs clarification and detail. I suppose with the utilization of a high density medium and based on the governing equations of magnetic levitation, the levitation height differences should become more pronounced even for minor differences in single cell density. Is this the reason that authors reach to the no density change conclusion?

In response to this comment, we have updated this part of the text: While we do not modulate the cell density and do not expect it to be affected as a result of sodium nitrite treatment (which should only modulate magnetic susceptibility), we cannot independently measure or verify that density remains unchanged, so we have now removed this statement from the text.

7) Authors can translate their levitation height measurements into single cell density values and compare these findings to the data presented in Figure 2c. Such a comparison would improve the foundation of the study.

Thank you for suggesting this idea. While we considered doing this to bring us back to the simulated levitation height predictions in Fig. 2, it would require making significant assumptions about the other variables that we cannot measure independently, especially the magnetic susceptibility of the cells. The lack of precise understanding and measurement of these individual variables, and the difficulty in sensitive separation, is why we pursued a multiparametric approach to this problem overall.

8) Statistical significance between groups is denoted with *, **, or ***, but corresponding p values should be specified in the figure legends. This is present for figure 3 but missing in figure 2.

Thank you for pointing this out, these have now been included in the figure caption for Figure 2 as well:

“The following labelling convention was used throughout the figures: * = $p < 0.05$; ** = $p < 0.01$; *** = $p < 0.001$; **** = $p < 0.0001$.”

9) Based on the other levitation images, Figure 3b lower boundary does not look like 0.

In this case, the cell levitation heights did occur near the bottom of the capillary (i.e. $h = 0$), as occurred with some samples (this variation is addressed in the Discussion, on page 10, line 17 onwards). This particular image was used as a visually clear example of the distribution, to demonstrate the relative height differences between the parasite-infected RBCs and uninfected RBCs within the same population, at higher magnification. This shows that parasitized cells demonstrate different behavior compared to their non-parasitized counterparts, regardless of the overall height of the population.

10) For Figure 3d and 3e, is there a reason for not including the microscope images of late stage iRBCs?

We have now included these images to be comprehensive (as below), following this comment.

Figure 3: Levitation of *P. falciparum* infected human erythrocytes from 3D7 cultures.

h refers to levitation height (μm), X refers to magnetic susceptibility (unitless), and ρ refers to density (g/mL).

(d) Levitation images of **(i)** uninfected RBCs, **(ii)** ring-stage parasitized RBCs (purified with streptolysin O; SLO), and **(iii)** mature-stage parasitized RBCs (purified with streptolysin O; SLO) in low- X , low- ρ medium. Scale bar = 200 μm .

(e) Levitation images of **(i)** uninfected RBCs, **(ii)** ring-stage parasitized RBCs (SLO-purified), and **(iii)** mature-stage parasitized RBCs (SLO-purified) in low- X , high- ρ medium. Scale bar = 200 μm .

11) For figure 3c, though it is obvious, the difference in the levitation height of parasitized cells should be depicted with a statistical test for completion. Also, perhaps the choice of bar colors can reflect that the first bar is a combination of the last two. And finally, the fraction range of cells can be depicted over the last two bars to help narration.

We have followed the recommended formatting advice for Figure 3c, using a scatterplot to better show the differences in levitation height for cell subpopulations in the same sample (with a different color used for the data series of each sample). This helps to emphasize that within each sample, parasitized cells always tend to levitate higher than their non-parasitized counterparts, even though the result of conducting a Welch's t-test was a p-value = 0.5. We have also added the fraction range of the cells to help with narration, as recommended.

12) In the caption of Figure 3, “For both (D)(iii) and (E)(iii)” should be corrected as “For both (d)(iii) and (e)(iii)” with lower cases. Furthermore, Figure 4.(aa) should be Figure 4.(a) These errors in the figure captions have been corrected following the feedback given.

13) Linear discriminant analysis is a neat way of dimensional reduction. However, it is not clear why authors did not try more straightforward approaches such as coefficient of variation or feature scaling with their distributions. Furthermore, authors should mention the selection criteria of LDA compared to other dimensional reduction protocols such as principle component analysis. Thank you for this thoughtful comment. We considered various approaches towards achieving our goal, which was to classify the samples into groups on the basis of multiple features. This is, of course, a common problem, and one for which myriad techniques exist. We chose linear discriminant analysis (LDA) because it was the most straightforward and fitting solution for our goal specifically. This is not only because it is a standard and long-standing method in the field for dimensionality reduction, but because it includes feature scaling, and most importantly because it is well-suited for performing classification. Principal component analysis, which is similar in many ways and often compared to LDA, is an unsupervised learning technique and

aims to maximize variance between data points along the projected axis(/axes), and it is excellent for dimensionality reduction. LDA, on the other hand, is a supervised learning technique, and while it is also effective for dimensionality reduction, it aims to maximize class separation (by maximizing variation between classes while minimizing intraclass variance), and is therefore excellent for a classification problem, which was exactly our goal. In summary, LDA not only includes feature scaling, but also uses class labels (i.e. supervised learning, with known values of ‘uninfected’ and ‘infected’ used in training) to strive for the best classification of samples into these categories. We have added more explanation to the Methods section, on page 16, line 5 onwards.

14) Authors should mention the final concentration of the gadolinium agent used for levitation in the text for reproducibility. Furthermore, a simulation mapping of medium density vs. the gadolinium concentration for the levitation height (or single cell density) difference of parasitic cells may offer insight to improve the sensitivity of the test.

We have included this detail in the manuscript under “General preparation of levitation mixtures for experiments” in the Methods, on page 13, line 5 onwards (we use a final concentration of 30 μM). The results of a simulation mapping medium density versus cell levitation heights, and gadolinium concentration vs. cell levitation heights, can be found in Fig. 2e.

Figure 2. Levitation changes from independent variations in density and magnetic susceptibility, and theoretical predictions of levitation heights of cell types under different conditions, based on reported and estimated values of cell densities and magnetic susceptibilities.

h refers to levitation height (μm), X refers to magnetic susceptibility (unitless), and ρ refers to density (g/mL). “RBC” refers to red blood cells, or erythrocytes; “uRBC” refers to uninfected RBCs; “ring-iRBC” refers to ring-stage synchronized infected RBCs, and “troph-iRBC” refers to trophozoite-stage (i.e. more mature) infected RBCs.

(e)(i) Predicted heights of ring-stage and uninfected RBCs at varying medium densities, showing increased separation in a high- ρ medium, marked in a yellow box (calculated at low medium $X = 1.12 \cdot 10^4$). **(ii)** Predicted heights of ring-stage and uninfected RBCs at varying medium magnetic susceptibilities, showing increased separation in a low X medium, marked in a yellow box (calculated at high medium $\rho = 1.11$ g/mL).

Reviewer #3 (Remarks to the Author):

Deshmukh and colleagues describe a multiparametric approach to profile cells based on minute differences in density and magnetic susceptibility. The malaria parasite *Plasmodium falciparum* is used to demonstrate the feasibility of this approach as infection of red blood cells with this pathogen simultaneously decreases the density of the host cell and increases its magnetic susceptibility. The minute differences between ring-infected and uninfected red blood cells in terms of density and magnetic characteristics have complicated separation by means of a single parameter. The authors show that a multiparametric approach combining these parameters allows a better separation of these stages.

The manuscript is well-written and organized in a logical manner. The findings described are novel and could be of interest to a broader audience. However, there are some aspects of the paper that merit attention and require clarification.

Thank you for this detailed and thoughtful feedback that has added a new perspective to important questions we have been considering around this research work, as well as very specific suggestions and new ideas that have added a lot of value as well. We hope we have been able to improve the manuscript by following your recommendations.

Major comments

1. One aspect that merits attention is the application of the platform. In the introduction, from lines 48 onwards, the authors propose the use of their newly developed platform in resource-poor settings. They further elaborate on this in the discussion (lines 29-41). My concern here is that the authors have not used any patient blood in the work described here and thus this claim cannot be substantiated. For this it would be useful to show data with patient derived material, or as a proxy, with parasites diluted in whole blood. It appears that figure S8 shows some preliminary data on this, but it is not discussed in the main text. Validation with patient blood is something that is very important to our work, and we believe it is something that will take further work and care to develop a suitable solution for this from the starting point presented in this paper. Thus, while we focused on lab cultures for this portion of the work, we will primarily be focusing on patient sample testing in our continuing work in the same research path, since there are important adaptations that needed to be made to this underlying technology to ensure suitability and fit for whole blood patient samples and resource-limited settings. We have conducted some very preliminary testing with a limited number of clinical samples using the same procedure presented in this paper, as shown below.

Testing levitation protocol and classification algorithm performance beyond standard negative healthy controls and high-parasitemia positive samples to *ex vivo* whole blood samples freshly collected from clinical malaria patients diagnosed with *P. falciparum* malaria. These samples were tested at a clinic in eastern Uganda with an area with malaria transmission, and images were taken on a portable imaging platform with the same levitation devices and protocols as used for the results in Fig. 4. Figure shows LDA score outputs from the same algorithm developed for results in Fig. 4, applied to *ex vivo* samples levitated in the same low-density medium protocol, with $n = 7$ samples tested. 6 out of 7 samples were correctly classified as malaria-positive.

Furthermore, it is not clear to me what the advantage is of the levitation method in resource constrained communities over the already available simple methods such as the use of giemsa stained blood films/dipstick. While the platform described is of interest as a fundamental biological finding, questions remain as to what the authors see as a realistic application. The authors do some suggestions on this in lines 5 and 6, but how this is done and how this would provide advantages over established methods is unclear.

We aimed to shed more light on the biophysical transformation occurring in *P. falciparum*-infected cells, a topic of significant research focus over the decades, and the nexus of the devastating symptoms and global health effects of malaria. We were able to show that ring-stage infected cells, the primary stage that flows freely in patient circulation, does show early signs of this biophysical transformation that appears more readily in later stages. This procedure can be conducted in comparable or less-resourced settings than common biophysical separation methods such as Percoll gradient separation (which uses a centrifuge) and magnetic columns (which can be costly) that are the standard in a wide range of laboratory applications from basic to clinical science. By presenting a technique that can be used in a range of operative settings including many that are more accessible in malaria-endemic regions, and enabling manipulation of the ring-stage, we hope to form the foundation for new types of scientific experiments and clinical investigations that were not possible before due to technical limitations. We also hope that this platform forms the foundation for more patient-interfacing (i.e. point-of-care) testing subsequent to further technological modifications.

We have also modified the 4th and 5th paragraphs of our Discussion to reflect this more clearly (page 9, line 26 onwards and line 42 onwards).

2. Fig S5 addresses some variables that can result in different levitation patterns. This highlights the importance of using uninfected RBCs that are treated exactly the same as the infected RBCs (so kept in culture for similar periods of time as the infected cells and sorbitol treated). Given that the authors suggest to use the platform in resource poor settings, where it may not be possible to use fluorescence for parasite detection, I strongly suggest that the authors show a side by side comparison of uRBC, RBC that have been treated exactly as the iRBC and the iRBC (preferably using whole blood).

This is very insightful and is exactly what inspired us to run this experiment to investigate the conditions and the infection itself on the biophysical signature of the cells). The data shown in Fig. S5 (reproduced below) is comparing these conditions, by measuring side-by-side the levitation heights of uninfected RBCs with and without exposure to culture conditions, to those of infected RBCs under culture conditions. While this would be difficult to do with whole blood since culturing iRBCs is typically done after removal of the immune cells/leukocyte components in order to enable infection, we have explored this with RBC samples in Fig. S5 via exposure to culture media, exposure to the same incubation conditions, and finally sorbitol exposure as well.

Figure S5: Comparison of control (non-infected) RBCs in normal storage (4 degrees Celsius) and in culture conditions (incubated with culture media, and one additionally sorbitol-treated) for four days.

3. Another important potential application is the opportunity to isolate infected young ring stage parasites. It is not clear whether this is already feasible or whether the authors consider this to be possible in the future. For example, in figure 1A a picture is shown of a blood smear after levitation to equilibrium. Does this mean that the parasites can be isolated from the device? Please clarify. It would indeed be possible to extract cells from the capillary after levitation, and such a technique has been applied with other cell types in a flow-based levitation setup, for example in one research paper cited in our manuscript (Puluca, N. et al. Levitating Cells to Sort the Fit and the Fat. *Adv. Biosyst.* (2020) doi:10.1002/adbi.201900300). We could implement this kind of attachment to our system to extract cells upon separation at equilibrium, or in flow. To clarify here, the blood smear is made with an aliquot of infected blood before levitation, but it would be possible to do the same with an aliquot of blood levitated in the capillary and subsequently extracted simply by use of a pipette tip, or by flowing the cells through with the use of tubing and syringe pumps to regulate flow.

On p8 line 5 the authors mention the isolation of specific stages responsible for transmission. Can the authors please be more specific about this? How is this done and what do the authors want to achieve?

This is a future application we would like to explore, by aiming to detect and isolate gametocytes, the sexual stage of the parasite that is present in circulation in very small numbers, and it is important to study in populations in endemic regions because they are responsible for human-to-mosquito transmission. Currently, researchers use magnetic separation techniques such as MACS columns to try to isolate gametocytes from blood samples, but a technique that is more sensitive to these very rare cells, and also one that is less expensive and dependent on electricity, would be an aid to researchers studying gametocytes in the field. We would anticipate testing our technique for the isolation of gametocytes by designing the device parameters to take advantage of the uniquely high magnetic susceptibility in this stage, as well as tuning it to the density of those cells since we can take advantage of multiparametric separation. We could then extract and isolate the specific cell type of interest by using a flow-based separation technique as mentioned above.

In lines 27 (p8) the authors mention that molecular analyses are still possible because the cells are intact and viable. This suggest that the authors have purified the cells and checked their viability. Have the authors done this? If this can be done it is important to show the data on this. Viability in the context of RBCs has a different significance in comparison to many other mammalian cell types, so we mainly intend to describe that cells remain alive and undamaged after going through levitation. Since so many important characteristics of RBC are biophysical, one can estimate the status of these cells by looking at RBC morphology, rheology, membrane porosity, etc. We have found that when these cells are damaged or dying, their membrane porosity increases significantly, and they are no longer able to levitate and fall out of sight of the capillary. Thus, we have found RBCs to remain intact and with their normal morphology during out experiments. While we have not tested viability in the context of *P. falciparum*-infected cultured cells specifically and therefore have updated the text to be clear about this, we have investigated this in other cell culture lines and therefore would hypothesize these cells to be viable after brief exposure to the levitation conditions. More importantly, for molecular analyses, we wanted to indicate that our procedure does not inflict any kind of significant molecular manipulation on the sample.

Minor comments

1. Figure 1C: since the density of infected RBCs is slightly lower, it may be good to show a minor difference between uRBC and ring-RBC on the ρ axis (they now look the same).

Thank you, we have now corrected this in the figure.

2. In the results section, p3 starting at line 47: it is stated that pathological conditions may alter the magnetic signature of RBCs. Given that malaria-infection patients often suffer from anemia wouldn't this complicate the levitation read-out of patient derived infected cells?

This is an important insight into the advantages and limitations of a label-free biomarker such as levitation height. As a consequence of biophysical features and changes, it can reflect various pathologies or states of the health spectrum. Since anemia is an important consequence of not only malaria but various other illnesses, this is something we consider to be an important relevant factor. To explore this, we have conducted some preliminary experiments in the case of sickle-cell anemia, an inheritable condition that correlates highly with populations living in areas that

have long been malaria-endemic, and is thus very relevant to any studies around malaria. Conversely to the effect of malaria on RBC levitation height (increasing height, due to lower density and higher magnetic susceptibility), sickle cell anemia was found to decrease levitation height, hypothesized to be due to the dehydration phenotype observed in many affected cells that leads to increased density and thus should map to a lower levitation height (we cite this existing work in the final paragraph of our Discussion: Knowlton, S. M. et al. Sickle cell detection using a smartphone. *Sci. Rep.* (2015) doi:10.1038/srep15022). While doing an in-depth experimental investigation into various types of anemia was not within the scope of this study, it is certainly of importance and relevance, and something that we hope to explore further in future work. This is another consideration for adding a dimension of labelling, to increase specificity amidst a variety of causes that influence cellular biophysical characteristics. One example of this is fluorescent staining which is already compatible with this system.

3. p5, line 3 (Figure 2C): levitation heights are not shown in this figure.

Thank you for pointing this out, we included the missing figure in the supplementary information now as Fig. S10 (and have updated the main text to correctly refer to this now).

Fig. S10. Predicted heights of erythrocytes of different infection stages in low-density medium (4 µm separation between ring-stage and uninfected) and high-density medium (23 µm separation between ring-stage and uninfected) under a fixed set of medium conditions other than density.

4. p5, line 10. The ref number appears incorrect (possibly this should be ref 35?)

We have updated this to the correct reference now, thank you for pointing this out.

5. Figure S3: while panel c shows a simulation of a 1:10 dilution, there are many more dots (cells) depicted than in the undiluted sample (panel a). Can the authors clarify this? Also, the figure refers to Fig 2F, but this is not present. Please correct.

We have now clarified this further in the figure caption, panel a is supposed to refer to a cell mixture at 1:100 dilution, while panel b is more dilute (1:1000) and panel c is less dilute than in a (1:10). We hope the language is more clear now. We have also corrected the figure number referenced in the caption.

Updated figure caption:

“Figure S3. (A) Simulation of ring-stage synchronized infected culture at 5% parasitemia in a high-density medium (1.11 g/mL) at low magnetic susceptibility ($1.12 \cdot 10^4$), as in Fig. 2E, at a

standard dilution factor (1:100). **(B)** Simulation of the same sample and conditions, but more dilute (1:1000). **(C)** Simulation of the same sample and conditions, but less dilute (1:10).“

6. In figures 2 and 3, bar charts are used. It is not noted what the error bars represent. Scatter plots would be preferred because this gives a better sense of the variation between measurements.

Thank you for this suggestion, we have now shown scatter plots overlaid on box and whisker plots wherever applicable for greater clarity on the data and its inherent variation. For example, in **Figs. 2a-c**, and **Figs. 3d-e**:

(portion of Figure 2 below)

(portion of Figure 3 below)

7. p6, line6. In the methods section sorbitol synchronization is described, but it is not clear what the age of the rings is. How tight was the synchronization? ‘Old’ rings may give different results than very ‘young’ rings.

Sorbitol synchronization results was checked with microscopic examination of Field-stained smears to check how synchronous the parasites were. However, although within our experiments we conducted levitation with some samples that were taken shortly after bursting and synchronization, we did not strictly control the age of the rings (hours post-synchronization, nor hours post-bursting) as long as it was within 18 hours, so this is a limitation of our study that we would aim to improve on in future work. We agree that the age of the rings is important, as we could expect the effects we observe to scale with the age of the rings, and it would be significant

to investigate the threshold of separability in ring “age”. We have added further detail to the Methods section describing synchronization, on page 14, line 9 onwards.

8. In figure 3 and throughout the manuscript the term ‘healthy’ is used (for example also on p7 line6). This suggests that the authors have checked the health status of the cells involved. This is confusing; please use uninfected instead.

Thank you for pointing this out, we have now modified our language throughout the manuscript to use the more accurate term “uninfected” instead of “healthy”.

9. Figure 3: Does the acridine orange have any effect on the levitation?

We do not expect that acridine orange should not have any significant effect on levitation, because it is not considered to be a ferromagnetic or paramagnetic substance and thus would not significantly change the magnetic susceptibility of the medium. It is also used in very low concentrations so it should not significantly change the parameters of the medium as we prepare it. We did conduct a control experiment on RBCs with and without acridine orange staining (at the standard concentration used in other experiments) and found no significant difference between the levitation heights of RBCs incubated with acridine orange, or PBS as a control.

Fig. S11. Acridine orange incubation (at standard concentrations used for RBC staining as per the protocol outlined in the Methods, in the standard medium and levitation time) does not have a significant effect on levitation height of RBCs (uninfected), $p = 0.605$. Control, with PBS ($n = 3$), and acridine orange-treated ($n = 3$) samples were compared.

10. In figure 3 a(i) many more cells appear to be present than in figure 3 a(ii). Please comment. Also, the uninfected cells (not fluorescent) have a different levitation height than in (i); can the authors explain?

The appearance of more cells in Figure 3a(i) may be due to the wider spread of cell heights in a(ii) and also because of the 3-dimensional nature of the cell levitation, which results in cells being stacked in front of or behind each other when imaged horizontally at one plane of focus,

rendering some variation in the number of visible cells in any one given plane of focus compared to other samples.

Regarding the uninfected (non-parasitized) cells in the infected population (a(ii)), this variance is in line with what we have observed throughout our experiments, as we explore in depth in the Discussion (page 10, line 17 onwards). We hypothesize that there may be some effect of culture conditions on the cells (as seen in Fig. S5), or some extracellular communication or ageing effect from the infection, that affects all cells in the infected population, not just the parasitized ones. Small differences in the global levitation profile of the whole population such as this one may also be due to individual variations in the biophysical characteristics of the donor's RBCs, or due to experimental variation. Hence, we look at patterns in the height distribution itself (specifically, the effects induced by the parasitized components and how they differ from the non-parasitized counterparts), rather than just the mean height of the whole band, to determine the signature of infection.

11. Figure 3 d and e: it is important to include SLO treated uRBC controls as well, because from figure S4b it appears that uninfected cells in a normal culture differ in levitation from uninfected cells (fluorescence negative) in a SLO treated culture. Or is this a sorbitol effect? In d(ii): are this all parasites or is there also debris?

Figure S4 is indeed showing uninfected cells under culture conditions, but not under the effect of SLO, only the conditions used to support *P. falciparum* infection. The other condition shown is culture conditions (i.e. stored in the temperature and gas-controlled incubator in culture media) as well as sorbitol treatment, since that is applied to all infection-containing cultures that are synchronized to the ring stage in our experiments. This showed that the culture conditions, and then also sorbitol treatment, seems to have an effect on uninfected cells as discussed in depth in the Discussion, but this does not include any effect of SLO treatment. Due to SLO's effect of forming pores in cholesterol-rich membranes, its effect on uninfected RBCs would be to induce complete lysis. This is the mechanism that allows us to purify infected cultures into populations that contain only parasitized cells (in intact form), by lysing all the uninfected cells. Thus, it would not be possible to observe levitation characteristics of SLO-treated uRBC controls since the cells would no longer be intact. Finally, there is occasionally some debris present in samples, especially in lysed samples such as these SLO-treated samples.

12. P6 line 32. Mature-stage parasites are mentioned. Please explain how these stages were obtained (how synchronous were they?). It would be good to show a picture (at a higher magnification than that is used in fig S7) of the material that was used for the levitation (a stained slide with stained parasites).

Mature stages are obtained post-synchronization for rings after being cultured for another 24-32 hours (before bursting and beginning the next cycle as rings). Mature stages were checked for synchronicity by microscopy again. We have included here an example of the SLO-purified iRBCs from a mature stage (trophozoite stage) with acridine orange staining of nucleic acids under fluorescence microscopy to show the degree of purification.

13. P6 line 48. Rather than using 'disease' state I would suggest to use 'infection' state, because disease is not measured here.

Thank you for pointing this out, we have corrected the wording as suggested.

14. P7 line 14. The authors summarize different features of infected and uninfected blood samples. Would the authors consider blood group to be a variable? And what about the temperature of the sample? Could this influence the measurement?

While blood group is certainly an important variable when it comes to RBCs/erythrocytes in general, when it comes to bulk biophysical effects on the cell as a whole, we don't expect the effect of these molecules presented on the cell membrane to have a significant effect on the individual cells' levitation height in a measurable way. However, we would need to test this experimentally to see what effect blood group may have (we did not have access to this information with our samples since any information about the donor was removed and de-identified before reaching us).

Temperature is an important physical variable with universal effects, and certainly one that we expect to be relevant to our studies, and we have conducted some preliminary experimentation in this space. We hope to continue working on this question and be able to share any interesting results that come up. One important aspect of temperature that we have focused on is its effect on the paramagnetic susceptibility of the sample. Since thermal vibrations affect the dipoles, paramagnetic susceptibility is inversely proportional to temperature. The Curie temperature (the transition point above which materials lose their paramagnetic nature) for most elements is far from room temperature, but gadolinium happens to have a Curie temperature very close to room

temperature (292 K) which suggests that small shifts in the temperature range at which we conduct our experiments could theoretically affect the paramagnetic susceptibility of the medium in which we are conducting our experiments. Since we adjust and fix the magnetic susceptibility of the medium as a controlled variable, any effects from changing temperature are thus important to consider. While these experiments are ongoing, this is another angle of exploration that has come out of this work. This is also relevant to consider in the context where this platform would be translated to a variety of operative settings, perhaps in different climates and without steady access to temperature control.

REVIEWERS' COMMENTS:

Reviewer #1 (Remarks to the Author):

The manuscript "Microscale magnetic levitation is used to combine minute density and magnetic susceptibility differences to enhance biophysical separation of cells" by Deshmukh et al. describes an setup to discriminate red blood cells infected by plasmodium falciparum.

The revised version of the manuscript is well written. I have only some minor suggestions:

1. On page 2,3 - references are not displayed correctly.
2. On page 5 - Line 25 and 26, repetition of : We used...
3. On page 9, line 37 - I do not understand the statement that low density medium does not use susceptibility? Is this not the main point of the manuscript.
4. On page 14 - It is very minor, but the expression "bending light" should be replaced by reflecting.

Reviewer #2 (Remarks to the Author):

My concerns were largely addressed.

Reviewer #3 (Remarks to the Author):

I would like to thank the authors for carefully addressing all of the points that have been brought up.

Overall, the authors have adequately addressed my concerns.

We thank the reviewers for the time taken to review our revised manuscript in detail and to provide thoughtful and constructive feedback that has led to the improvement of the paper overall. Below, we have described how we have responded to each of the comments left by the reviewers in their last review. We have also taken the time to do extensive proofreading and improved the details of the manuscript and supplement, especially in the methods, and made the formatting of the figures more consistent.

Reviewers' comments:	Response:
Reviewer #1 (Remarks to the Author): The manuscript "Microscale magnetic levitation is used to combine minute density and magnetic susceptibility differences to enhance biophysical separation of cells" by Deshmukh et al. describes an setup to discriminate red blood cells infected by plasmodium falciparum. The revised version of the manuscript is well written. I have only some minor suggestions:	Thank you for these comments. Responses to the specific suggestions are below.
1. On page 2,3 - references are not displayed correctly.	The reference formatting in this location has been corrected.
2. On page 5 – Line 25 and 26, repetition of : We used...	This repetition in the sentence has been removed.
3. On page 9, line 37 - I do not understand the statement that low density medium does not use susceptibility? Is this not the main point of the manuscript.	We have clarified this sentence since it was unclear. It now reads as: "As hypothesized, ring-stage separation performed better in high-density medium than in low-density medium. This is because the high-density protocol takes advantage of both density and magnetic susceptibility differences in the cell, while the low-density protocol primarily depends on only the cell's density differences to induce separation." The message here we are trying to convey is that the high-density medium protocol was designed to take better advantage of not only density changes in the cells, but also take greater effect from the magnetic susceptibility changes as well.
4. On page 14 - It is very minor, but the expression "bending light" should be replaced by reflecting.	We have corrected this language accordingly, and also added a new diagram in the supplement (Figure S9) to visually describe the setup more clearly and accessibly.
Reviewer #2 (Remarks to the Author): My concerns were largely addressed.	Thank you for your time and feedback.
Reviewer #3 (Remarks to the Author): I would like to thank the authors for carefully addressing all of the points that have been brought up. Overall, the authors have adequately addressed my concerns.	Thank you for reading through and conveying this feedback.